# The role of highly oxygenated organic molecules in the Boreal aerosol-cloud-climate system

Pontus Roldin [1*], Mikael Ehn [2], Theo Kurtén[3], Tinja Olenius[4], Matti P. Rissanen[2], Nina Sarnela[2], Jonas Elm [5], Pekka Rantala[2], Liqing Hao[6], Noora Hyttinen [7], Liine Heikkinen [2], Douglas R. Worsnop[2,8], Lukas Pichelstorfer[2,9], Carlton Xavier [2], Petri Clusius[2], Emilie Öström[1], Tuukka Petäjä [2], Markku Kulmala[2], Hanna Vehkamäki [2], Annele Virtanen[6], Ilona Riipinen[4] & Michael Boy [2]

Over Boreal regions, monoterpenes emitted from the forest are the main precursors for secondary organic aerosol (SOA) formation and the primary driver of the growth of new aerosol particles to climatically important cloud condensation nuclei (CCN). Autoxidation of monoterpenes leads to rapid formation of Highly Oxygenated organic Molecules (HOM). We have developed the first model with near-explicit representation of atmospheric new particle formation (NPF) and HOM formation. The model can reproduce the observed NPF, HOM gas-phase composition and SOA formation over the Boreal forest. During the spring, HOM SOA formation increases the CCN concentration by ~10 % and causes a direct aerosol radiative forcing of $-0.10$ W/m$^2$. In contrast, NPF reduces the number of CCN at updraft velocities < 0.2 m/s, and causes a direct aerosol radiative forcing of $+0.15$ W/m$^2$. Hence, while HOM SOA contributes to climate cooling, NPF can result in climate warming over the Boreal forest.

[1] Division of Nuclear Physics, Department of Physics, Lund University, P. O. Box 118SE-221 00 Lund, Sweden. [2] Institute for Atmospheric and Earth System Research (physics), University of Helsinki, P.O. Box 64FI-00014 Helsinki, Finland. [3] Institute for Atmospheric and Earth System Research (chemistry), University of Helsinki, P.O. Box 64FI-00014 Helsinki, Finland. [4] Department of Environmental Science and Analytical Chemistry (ACES), Stockholm University, SE-106 91 Stockholm, Sweden. [5] Department of Chemistry and iClimate, Aarhus University, Langelandsgade 140, 8000 Aarhus C, Denmark. [6] Department of Applied Physics, University of Eastern Finland, P.O. Box 162770211 Kuopio, Finland. [7] Nano and Molecular Systems Research Unit, University of Oulu, P.O. Box 300090014 Oulu, Finland. [8] Aerodyne Research, Inc., Billerica, MA 01821, USA. [9] Division of Physics and Biophysics, Department of Materials Research and Physics, University of Salzburg, Hellbrunnerstrasse 34, 5020 Salzburg, Austria. *email: pontus.roldin@nuclear.lu.se

Atmospheric aerosol particles affect Earth's radiation balance by scattering and absorbing solar radiation as well as by acting as cloud condensation nuclei (CCN) and thereby alter the optical properties and lifetime of clouds. The impact of volatile organic compounds (VOC) on the climate is still highly uncertain because of incomplete fundamental understanding of VOC emission rates, the oxidation of VOC and the contribution of the formed oxidation products to aerosol particle formation, particularly the growth of newly formed particles (1–2 nm in diameter) to CCN[1–3]. In order for an organic vapour to contribute to the initial growth of new particles it needs to be an extremely low-volatility organic compound (ELVOC) or be very reactive at the particle surface or in the condensed phase[4–6]. When VOC react with oxidants, i.e. ozone ($O_3$), hydroxyl radicals (OH) or nitrate radicals ($NO_3$), peroxy radicals ($RO_2$) are formed[7]. A fraction of these $RO_2$ can autoxidize[8] and finally react with other radicals to form either closed shell monomers or dimers[9]. Based on experimental[8–19] and theoretical studies[8–11,18–22] autoxidation can lead to very rapid (sub-second to minute time scale) formation of highly oxygenated organic molecules (HOM), which are capable of driving initial nanoparticle growth[23]. HOM are formed efficiently from ozonolysis of monoterpenes with endocyclic double bonds (e.g. α-pinene and limonene) and also by monoterpenes oxidized by OH[9,10,17]. There is also experimental evidence that NO can suppress autoxidation[9]. However, until now we have been completely lacking a chemically sound but still computationally efficient mechanism that can be used in atmospheric chemistry transport models (CTMs) to simulate the contribution of HOM to the growth of new particles in the atmosphere.

In some earlier publications, the term HOM was used interchangeably with the term ELVOC, inferring saturation vapour pressures below $10^{-9}$ Pa ($\sim 2.5 \times 10^5$ molecules cm$^{-3}$)[9,10]. However, both functional group contribution methods[24,25] and quantum chemical calculations indicate that most HOM monomers with ≤10 carbon atoms are not ELVOC[6]. Here, we define HOM as molecules with at least six oxygen atoms formed from peroxy radical autoxidation of VOC[23,26]. Organonitrates, which are formed when $RO_2$ react with nitrogen oxide radicals (NO), can contain 6–7 oxygen atoms without involvement of peroxy radical autoxidation. Thus, in comparisons of modelled and observed total HOM organonitrate (HOM-$NO_3$) concentrations, only species with at least eight oxygen atoms are considered (see Methods for a more detailed motivation).

In this work we develop the first comprehensive peroxy radical autoxidation mechanism (PRAM)[27] for production of HOM from monoterpenes and couple it to the Master Chemical Mechanism (MCMv3.3.1)[7,28,29]. PRAM has the potential to become a widely used mechanism for more realistic representations of HOM SOA formation in regional and global scale CTMs. We demonstrate a very good agreement between the modelled and measured gas-phase HOM composition, secondary organic aerosol (SOA) mass concentrations and new particle formation (NPF) over the Boreal forest, and apply the new model to explore the climatic implications of HOM. According to our model simulations both HOM SOA formation and NPF substantial impact the Boreal aerosol-cloud climate system. However, while HOM is contributing to climate cooling by increasing the number of CCN and the magnitude of the direct aerosol radiative forcing, NPF can result in net climate warming by decreasing the direct aerosol radiative forcing and by supressing the cloud droplet number concentrations at low to moderate cloud updraft velocities.

## Results

**Smog chamber simulations**. The PRAM was constrained based on theoretical and experimental work on α-pinene

oxidation[9,12,17,22], but is generalizable to other monoterpenes as well since it takes into account that different monoterpenes have different HOM yields upon oxidation with ozone or OH (see Methods and Supplementary Table 1). Currently PRAM comprises 208 species and 1773 reactions (Supplementary Tables 2–4). The model used smog chamber specific HOM wall loss rates[9] (see Methods).

PRAM reproduces the observed total HOM, closed shell HOM monomers, HOM $RO_2$ and HOM dimer concentrations for a wide range of α-pinene+$O_3$ reaction rates, in the Jülich Plant Atmosphere Chamber (JPAC) (Fig. 1b, c). PRAM also captures the general patterns of the observed HOM mass spectrum (Fig. 1a). For most individual HOM monomer species (molecular masses in the range of ~230 u to ~380 u), the modelled concentrations are within 30% of the observed values. PRAM uses temperature dependent peroxy radical autoxidation reaction rates estimated based on quantum chemical calculations[14] (see Methods), which leads to lower HOM yields at low temperatures. At 270 K, the modelled HOM molar yield from α-pinene ozonolysis is only ~2%, while at 289 K the yield is ~7%, in agreement with the observations from JPAC[9] (Fig. 1b). The results are consistent also with α-pinene ozonolysis experiments in the CLOUD chamber in CERN, for which a HOM molar yield of 3.2% at 278 K has been reported[30]. For these conditions the HOM molar yield in PRAM is ~4%. During daytime, NO has a profound influence on the $RO_2$ chemistry and particularly on the HOM composition. This is considered in PRAM, which was evaluated against the observed HOM concentrations during α-pinene ozonolysis experiments with variable NO concentrations (Supplementary Fig. 1). PRAM and the observations in JPAC give a ~30% reduction in the total HOM(g) concentration when the NO concentration increases from 0 to 1 ppb$_v$, mainly attributed to the loss of HOM dimers. According to the observations and PRAM, ~25% of the HOM monomers formed from the $RO_2$+NO reactions are organonitrates. It has to be noted that in the atmosphere, night-time organonitrate HOM formation is also attributed to monoterpenes reacting with $NO_3$ (ref. [31]). In the JPAC experiments, however, the fraction of α-pinene reacting with $NO_3$ was always less than 6% (Supplementary Fig. 2). Therefore, due to limited experimental constraints, the current version of PRAM does not take into account HOM formation via monoterpene+$NO_3$ reactions.

To evaluate the volatility of HOM, as well as SOA formation from HOM, we implemented PRAM in the aerosol dynamics model for laboratory chamber studies (ADCHAM)[32] and simulated SOA formation during an α-pinene ozonolysis experiment with ammonium sulfate (AS) seed particle addition[9] (see Methods). According to the model simulations, ~50% of the SOA mass is formed from condensing HOM species (Fig. 2a).

The HOM dimers are most likely ELVOC, as indicated by their pure liquid saturation vapour pressures ($p_0$) estimated with the functional group contribution method SIMPOL[24], using the estimated molecular properties assigned in the Supplementary Table 2. However, most HOM monomers with 6–8 oxygen atoms were most likely not ELVOC at the temperature of the experiment (289 K), e.g. SIMPOL predicts $p_0$ in the range $1.4 \times 10^{-4}$–$2.0 \times 10^{-8}$ Pa. Therefore, their uptake onto the seed particles are at least partly limited by their volatility. According to our model simulations this likely explains why the closed shell HOM dimer fraction decreased when the seed particles were added to the chamber (Fig. 2b). With $p_0$ estimated with SIMPOL the model captures the observed change in the HOM gas-phase concentrations upon seed particle addition (Fig. 2b, d). This indicates that the HOM $p_0$ for the least oxidized and most volatile HOM species (i.e. HOM with 6–8 oxygen atoms) are within the right order of magnitude in the model.

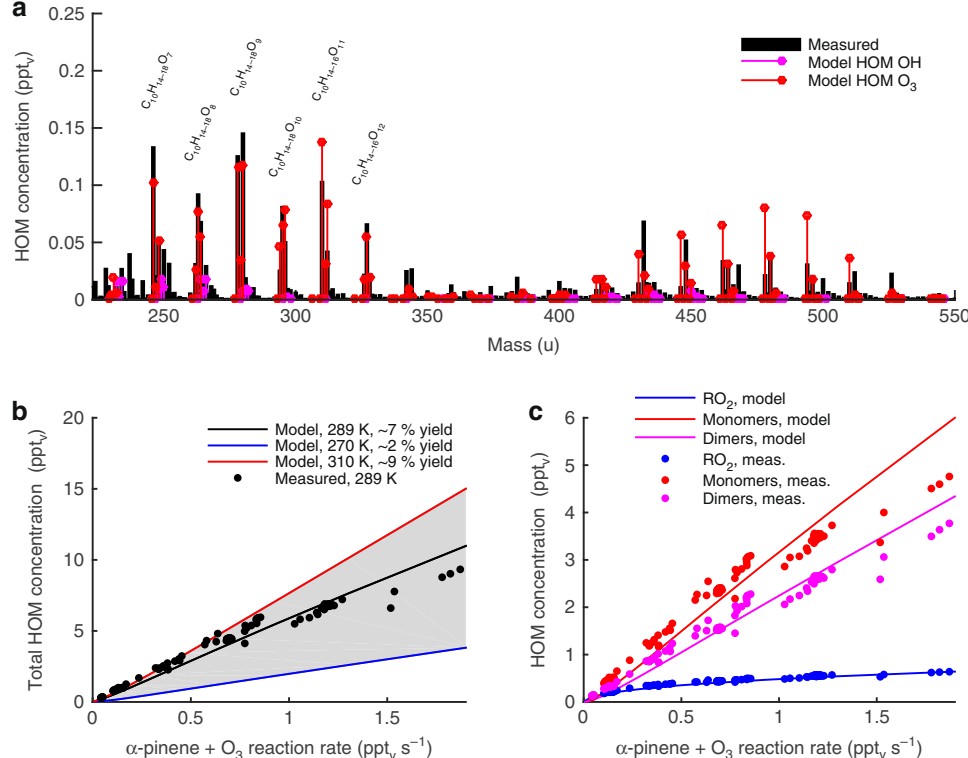

**Fig. 1** Highly oxygenated organic molecule (HOM) formation from $\alpha$-pinene. Modelled and measured HOM(g) concentrations during a JPAC $\alpha$-pinene ozonolysis experiment[9]. Panel **a** shows the modelled and measured HOM mass spectrum at an $\alpha$-pinene+$O_3$ reaction rate of ~0.3 $ppt_v$ $s^{-1}$. Panel **b** modelled and measured total HOM concentration at various $\alpha$-pinene+$O_3$ reaction rates. **c** Concentrations of HOM peroxy radicals ($RO_2$) and HOM closed shell monomers and dimers. In panel **a** the modelled HOM mass spectrum is shown separately for HOM species formed via ozonolysis of $\alpha$-pinene and via OH oxidation of $\alpha$-pinene. The mass of the reagent nitrate ion is not included in the measured molecular masses. The shaded area in **b** illustrates the variation of the modelled total HOM yield in the temperature range from 270 K to 310 K. The HOM were measured with a nitrate-ion-based chemical ionization atmospheric pressure-interface time-of-flight mass spectrometer (CI-APi-TOF) (see Methods)

Also the modelled SOA elemental composition is in agreement with the observations (Fig. 2c), although the model gives slightly higher hydrogen-to-carbon ratios (H:C). Both the modelled and measured oxygen-to-carbon ratios (O:C) of the total SOA mass are around 0.62, while the modelled O:C of the HOM SOA is 0.67.

**Sources and sinks of HOM in the atmosphere**. To evaluate PRAM for atmospheric conditions we implemented it into the chemistry transport model (ADCHEM)[33,34]. For this purpose ADCHEM was first set up as a stationary column model at the Station for Measuring Ecosystem-Atmosphere Relations II (SMEAR II) in Finland (61.85°N, 24.28°E) for the period 15–24 May 2013 (see Methods). Figure 3 compares the modelled and measured concentrations of HOM(g) $RO_2$, closed shell HOM(g) monomers without nitrate functional groups, HOM(g) dimers and closed shell HOM(g) monomers with nitrate functional groups (HOM-$NO_3$) at SMEAR II.

Table 1 summarizes the observed and modelled average HOM concentrations, the Pearson's correlation coefficients ($R$), the normalized mean bias (NMB) and the fraction of model predictions within a factor of two of the observations (FAC2). The HOM observations at SMEAR II have an estimated measurement uncertainty of approximately a factor of two[9]. Considering this, the modelled HOM concentrations are generally in good agreement with the observations. For the total and closed shell HOM monomers, the modelled concentrations are within a factor of two from the observations for 93% of the

time (Table 1). For the dimers, the correlation between the model and observations is high, but the model tends to underestimate the concentrations during the daytime (Fig. 3c). This is possibly due to other sources of highly oxygenated dimers during the daytime, which are currently not accounted for in PRAM. However, it should be noted that the measured dimer concentrations may also be influenced by contaminations, considering the low concentrations and how they were summed up (see Methods). The somewhat lower HOM-$NO_3$ concentrations in the model compared to the observations, especially during the night-time, may partly be attributed to missing HOM-$NO_3$ formation pathways via $NO_3$ oxidation of monoterpenes in the present version of PRAM.

The lowest correlation and highest normalized mean bias between the modelled and observed HOM species types are found for the peroxy radicals ($R = 0.60$, NMB $= 0.67$). These discrepancies may partly occur because the observed $RO_2$ concentrations in Fig. 3 represent the concentration of only 10 HOM species (see Methods). However, it is also possible that PRAM is missing some bimolecular and unimolecular $RO_2$ termination reactions, which is manifested when the mechanism is applied for atmospheric conditions, but not during the pure $\alpha$-pinene ozonolysis experiments in JPAC.

Ozonolysis and OH-oxidation of monoterpenes on average accounts for 79% and 21% of the modelled HOM production respectively. The HOM production via OH-oxidation of monoterpenes peaks around noon, above the canopy, where it accounts for ~40% of the total HOM production (Fig. 4b). However, inside the canopy the OH concentrations are substantially lower

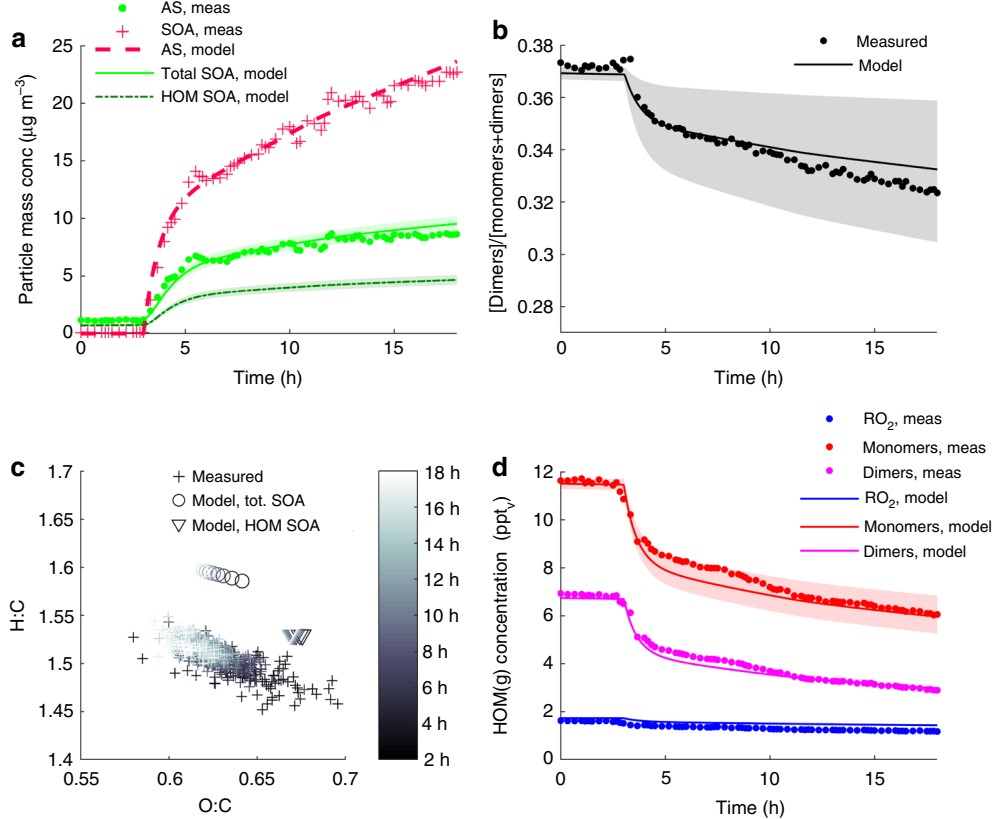

**Fig. 2** Highly oxygenated organic molecule (HOM) gas-particle partitioning. Model and measurement results from an $\alpha$-pinene ozonolysis experiment with ammonium sulfate (AS) seed particles[9]. Panel **a** shows the modelled and measured secondary organic aerosol (SOA) and AS seed particles mass concentration. **b** Relative dimer fraction of the total closed shell HOM gas-phase concentration. **c** SOA elemental composition, and **d** gas-phase concentrations of closed shell HOM monomers, dimers and peroxy radicals ($RO_2$). The shaded areas in panel **a**, **b** and **d** represent the range of model results obtained when the HOM pure liquid saturation vapour pressures were set to be one order of magnitude higher or lower than the values estimated with the functional group contribution method SIMPOL

(Supplementary Fig. 3) and OH-oxidation of monoterpenes only accounts for ~25% of the total HOM production around noon (Fig. 4a). In the canopy the HOM deposition losses are of the same magnitude as their condensation sink (Fig. 4a) and without considering HOM dry deposition losses, the model significantly overestimates the total HOM(g) concentration (Supplementary Fig. 4, Supplementary Table 5). This illustrates that dry deposition needs to be considered when comparing modelled and measured gas-phase HOM concentrations within the forest canopy. However, when integrated over the lowermost 2500 m of the atmosphere, dry deposition only accounts for 6.5% of the total HOM(g) losses (Fig. 4b). The high deposition losses of HOM(g) are a result of their generally low volatility and relatively high solubility in water, which in the model is described by their Henry's law coefficients (Supplementary Table 6, Supplementary Fig. 5). The modelled total HOM(g) concentration increases with altitude inside the canopy and reaches a maximum just above the top of the canopy, though the HOM(g) production is highest in the lowermost part of the canopy. This is because the condensation sink and dry deposition losses of HOM(g) are greater than their production rate inside the canopy, which causes a net downward flux of HOM(g) from the top of the canopy towards the surface (Supplementary Fig. 3).

**Formation and growth of new aerosol particles**. ADCHEM was also used to evaluate the impact of NPF and HOM on the aerosol particle population during spring 2013 (15–24 May) and spring 2014 (15 April to 5 May), for a total of 31 days. For this purpose

ADCHEM was run as a Lagrangian model along air mass trajectories arriving at SMEAR II with 3 h interval between each trajectory (see Methods).

According to the latest global CTM simulations, which use state-of-the-art NPF parameterizations from the CLOUD chamber in CERN[30,35], present day NPF can almost exclusively be explained by sulfuric acid ($H_2SO_4$) clustering with either ammonia ($NH_3$) or organic compounds formed from OH-oxidation of monoterpenes[36,37]. In the present work we considered NPF involving sulfuric acid clustering with both ammonia and organic molecules. The NPF through neutral and ion-induced clustering of $NH_3$ and $H_2SO_4$ molecules was modelled using the Atmospheric Cluster Dynamics Code (ACDC)[38], which was dynamically coupled to the aerosol dynamics model of ADCHEM (see Methods). This is the first time that a module that explicitly simulates time-dependent molecular cluster formation is implemented directly into an atmospheric CTM.

For the sulfuric acid–organics induced NPF we could not use an explicit representation of the cluster formation as the exact participating compounds remain elusive. Therefore, in order to include the possible contribution of the $H_2SO_4$–organics pathway, a semi-empirical parameterization was applied (see Methods). This parameterization has been shown to adequately represent the NPF rate of nanoparticles with a diameter of 1.5 nm ($J_{1.5}$) during real plant emission experiments in the JPAC smog chamber[39] and at the rural Boreal forest station Pallas in Northern Finland[34].

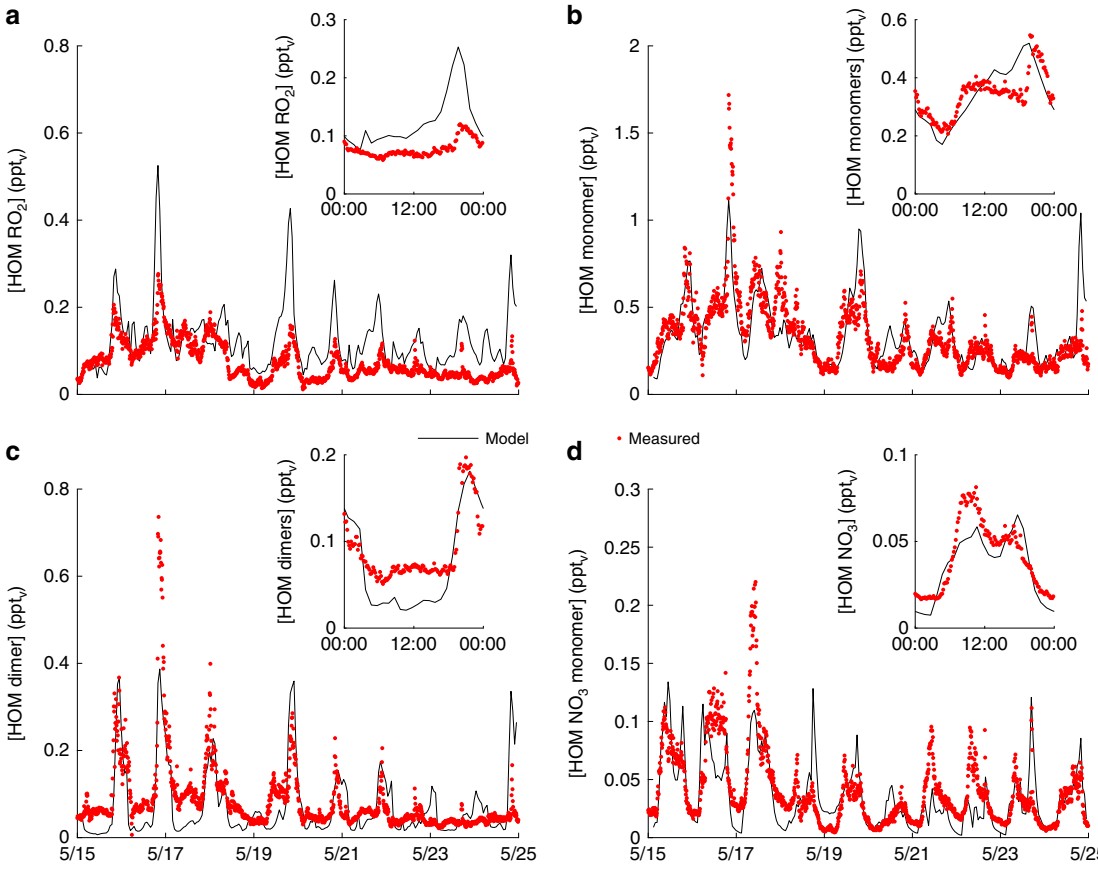

**Fig. 3** Highly oxygenated organic molecules (HOM) in the boreal forest. Modelled and measured HOM gas-phase concentrations at the Station for Measuring Ecosystem-Atmosphere Relations II (SMEAR II) between 15 and 24 May 2013. Panel **a** shows the concentrations of HOM peroxy radicals ($RO_2$), **b** HOM closed shell monomers without nitrate functional groups, **c** HOM dimers and **d** closed shell HOM organonitrate monomers (HOM-$NO_3$). The inset plots in each panel shows the mean diurnal concentration trends of each HOM species type

**Table 1 Evaluation of the modelled HOM concentrations at the Station for Measuring Ecosystem-Atmosphere Relations II (SMEAR II)**

| Species | $\overline{O}$ | $\overline{M}$ | $R$ | NMB (%) | FAC2 |
|---|---|---|---|---|---|
| Tot. HOM | 0.54 | 0.58 | 0.72 | 8 | 0.93 |
| Monomers | 0.33 | 0.34 | 0.72 | 3 | 0.93 |
| Dimers | 0.09 | 0.07 | 0.73 | −16 | 0.55 |
| HOM-$NO_3$ | 0.04 | 0.04 | 0.70 | −11 | 0.66 |
| HOM $RO_2$ | 0.08 | 0.13 | 0.60 | 67 | 0.61 |

Observed average HOM concentrations ($\overline{O}$) (ppt$_v$), modelled average HOM concentration ($\overline{M}$) (ppt$_v$), correlation coefficients ($R$), normalized mean bias (NMB) and the fraction of predictions within a factor of two of the observations (FAC2) at SMEAR II, 15–24 May 2013

As a complement to the control model run (CTRL), where we included all gas-phase chemistry and aerosol dynamics processes, we also performed four sensitivity test runs with: no HOM formation (NoHOM), no new particle formation (NoNPF), no new particle formation and no HOM formation (NoNPF-NoHOM), and only new particle formation via sulfuric acid clustering with ammonia. During spring 2013 the air masses reaching SMEAR II mainly originated from Russia and Eastern Finland, generally spending more than 2 days over the forest upwind SMEAR II. During spring 2014 the air masses mainly originated from the Arctic- and North Atlantic Ocean and spent less time over the forest upwind SMEAR II (Supplementary Figs. 6–8).

The biogenic VOC emissions along the trajectories were simulated using a modified version of MEGAN 2.04 (Model of Emissions of Gases and Aerosols for Nature)[40,41]. The modelled vertical monoterpene concentration profiles at SMEAR II are on average within ±20% of the observations at eight different altitudes between 4.2 and 125 m (Supplementary Fig. 9). The modelled concentrations of the relevant NPF precursors $H_2SO_4$ and $NH_3$ were compared with existing observations (Supplementary Fig. 10).

With the CTRL setup, ADCHEM predicts the general trends in the observed particle number size distributions at SMEAR II with reasonable accuracy (Fig. 5a, b) and the magnitude and diurnal trends in the number concentration of particles in the nucleation mode, Aitken mode, and accumulation mode (Fig. 5c, d, e). Both the modelled and observed particle number size distributions at SMEAR II show that the frequency and magnitude of the NPF events are substantially lower in May 2013 compared to April–May 2014. On average, the modelled NPF rate via clusters of $NH_3$ and $H_2SO_4$ contributes to 91% of the total number of new particles formed at SMEAR II respectively. The remaining 9% can be attributed to the NPF of organics with $H_2SO_4$.

The 2014 period is characterized by two strong NPF events, initiated by the formation of stable $NH_3$-$H_2SO_4$ clusters in the mornings on 16 and 23 April, followed by several days of apparent particle growth and increasing organic aerosol (OA) mass at SMEAR II (Figs. 5a, b and 6a). This apparent particle growth can to a large extent be explained by the gradually increasing time that the air masses spend over the forest upwind SMEAR II (Fig. 6a). The correlation coefficient between the

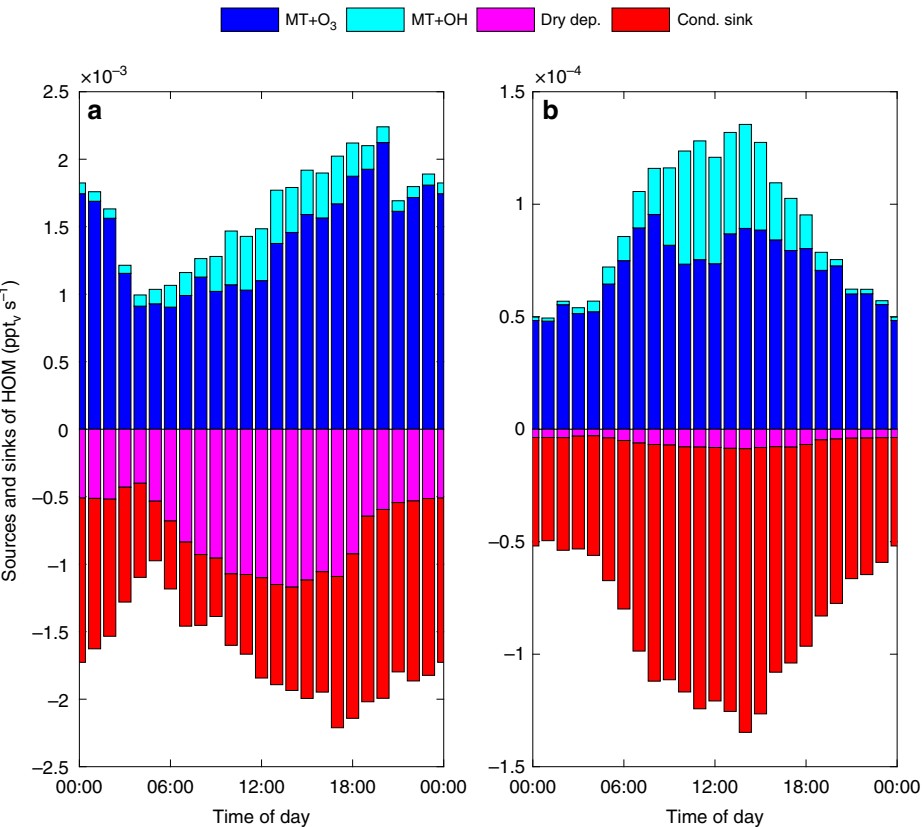

**Fig. 4** Sources and sinks of highly oxygenated organic molecules (HOM). Modelled median diurnal trends of the sources and sinks of HOM at the Station for Measuring Ecosystem-Atmosphere Relations II (SMEAR II) between 15 and 24 May 2013. Panel **a** shows the sources and sinks inside the forest canopy (0–18 m above ground) while panel **b** shows the integrated sources and sinks between 0 and 2500 m above the ground. Note the different scales on the y-axes. The panels show the sources of HOM(g) from monoterpenes (MT) oxidized with $O_3$ and OH and the sinks due to dry deposition and condensation onto aerosol particles

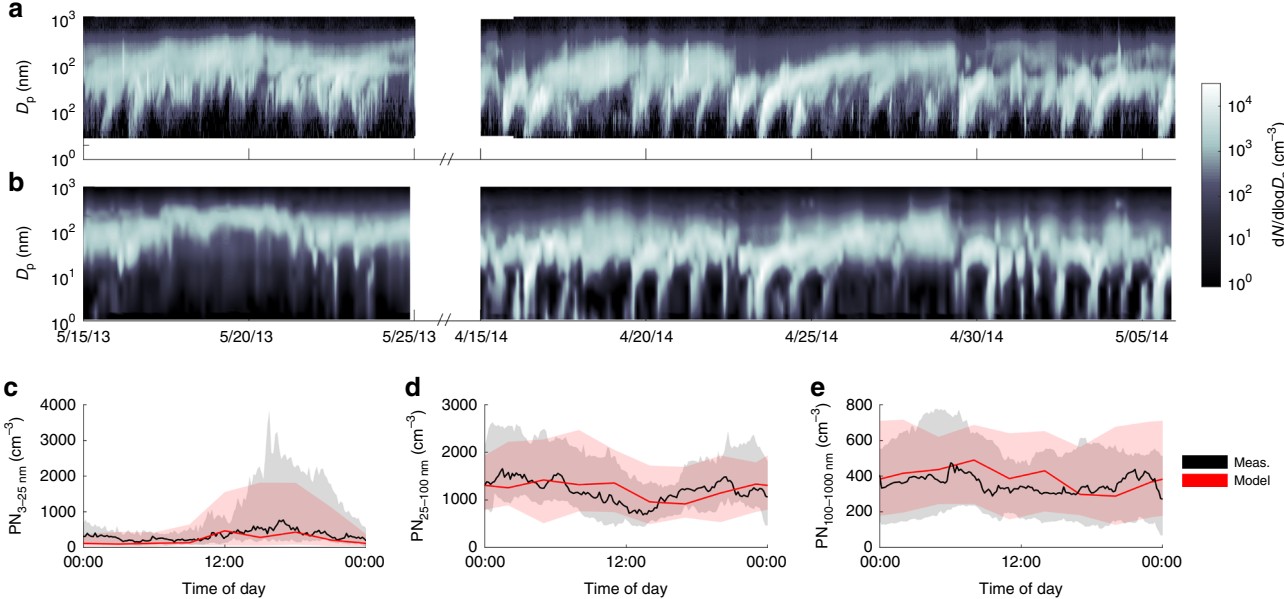

**Fig. 5** Aerosol particle number concentrations. Measured and modelled particle number concentrations at the Station for Measuring Ecosystem-Atmosphere Relations II (SMEAR II) from the periods 15–25 May 2013 and 15 April to 5 May 2014. Panels **a** and **b** show the measured and modelled particle number size distributions respectively. Panels **c**, **d** and **e** show the measured and modelled median diurnal cycles of particle number concentrations in the nucleation mode (3–25 nm in diameter, $PN_{3-25 \text{ nm}}$), Aitken mode (25–100 nm in diameter, $PN_{25-100 \text{ nm}}$), and accumulation mode (100–1000 nm in diameter, $PN_{100-1000 \text{ nm}}$) respectively. The shaded areas illustrate the measured (grey) and modelled (pink) ranges within the 25th and 75th percentiles. The particle number size distributions were measured with a differential mobility particle sizer (DMPS)

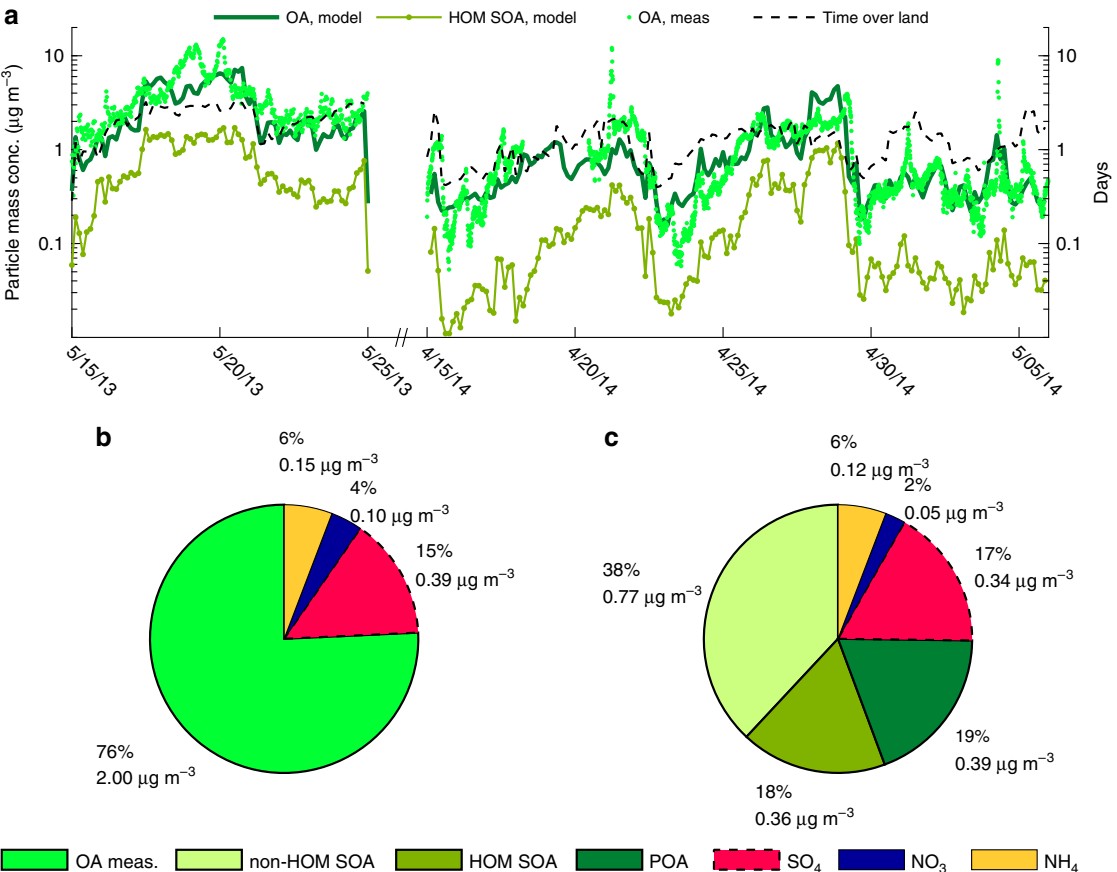

**Fig. 6** Aerosol particle chemical composition. Measured and modelled particle chemical composition at the Station for Measuring Ecosystem-Atmosphere Relations II (SMEAR II) station for the periods 15–24 May 2013 and 15 April to 5 May 2014. Panel **a** shows the measured and modelled submicron organic aerosol (OA) mass concentrations. Shown are also the modelled Highly Oxygenated organic Molecule Secondary Organic Aerosol (HOM SOA) mass and the number of days the air masses have spent over land upwind SMEAR II. Panels **b** and **c** show the average measured and modelled non-refractory submicron particle chemical composition, respectively. SOA accounts for 56% and the primary organic aerosol (POA) for 19% of the modelled submicron particle mass concentration respectively. The POA mainly originates from small-scale wood combustion and road traffic

---

**Table 2 Direct aerosol radiative forcing and CCN number concentration changes caused by HOM SOA and NPF**

| Process/species | $RF_{ARI}$ (W m$^{-2}$) | CCN change at $w = 0.1$ m s$^{-1}$ | CCN change at $w = 1.0$ m s$^{-1}$ |
|---|---|---|---|
| NPF | +0.148 | −10.4% | +32.0% |
| NPF($NH_3$-$H_2SO_4$) | +0.134 | −8.46% | +29.1% |
| HOM SOA | −0.100 | +2.9% | +11.5% |
| NPF & HOM SOA | −0.002 | −3.9% | +35.9% |

Modelled direct aerosol radiative forcing ($RF_{ARI}$) and relative change in the CCN number concentrations because of NPF, NPF formed exclusively from clustering of $NH_3$ and $H_2SO_4$, HOM SOA formation, and NPF and HOM SOA formation

modelled OA mass and the total time that the air masses have spent over land upwind SMEAR II during the past 4 days is 0.70. For the measured OA mass $R = 0.66$.

On average, HOM contribute to 18% of the modelled total submicron particle mass concentration (Fig. 6c) and to ≥20% of the particle mass in the size range from 1.5 to 200 nm in diameter. The non-HOM SOA (on average 38% of the submicron particle mass) mainly contributes to the mass in the accumulation mode (Supplementary Fig. 11). The modelled average aerosol particle composition is in close agreement with those measured with an aerosol chemical speciation monitor (ACSM) in 2013 and with an

aerosol mass spectrometer (AMS) in 2014, at SMEAR II (Fig. 6b, c). On average, the model yields 23% lower non-refractory submicron particle mass concentration than observed with the AMS and ACSM.

The modelled and measured O:C of the OA during the 2014 spring period are 0.70 and 0.72, respectively (Supplementary Fig. 12). The O:C of the modelled HOM SOA is on average 0.95. The higher O:C of the HOM SOA in the atmosphere compared to the smog chamber experiments (Fig. 2) is mainly due to smaller contribution of HOM dimers to the HOM SOA mass in the atmosphere (on average 9% compared to 38% during the smog chamber experiments). The reason for this is the lower concentration of $RO_2$ in the atmosphere compared to the JPAC experiments, which decreases the production of HOM dimers formed from $RO_2 + RO_2$ reactions.

**Impact of HOM and NPF on clouds and climate.** Finally, we use the modelled aerosol particle properties from the CTRL and the different sensitivity test runs to estimate the impact of HOM and NPF on the number of CCN and the shortwave radiation balance over the Boreal forest. We used the modelled particle number size distributions and size resolved particle chemical composition in the surface layer at SMEAR II as input to an adiabatic cloud parcel model[42,43]. We calculated the number of aerosol particles that activated into cloud droplets at different updraft velocities ($w$) (see Methods and Supplementary Fig. 13).

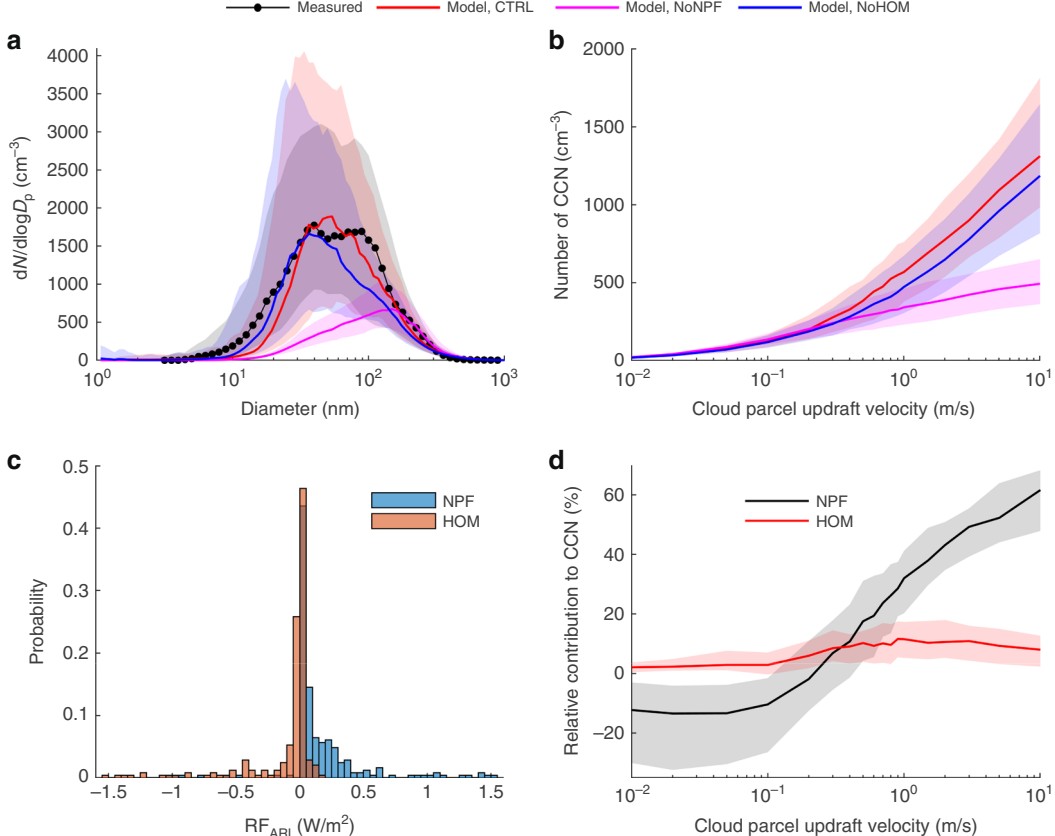

**Fig. 7** Climate implications of highly oxygenated organic molecules (HOM). Model and measurement results covering the conditions during the periods 15–24 May 2013 and 15 April to 5 May 2014 at the Station for Measuring Ecosystem-Atmosphere Relations II (SMEAR II). Panel **a** shows the measured and modelled median particle number size distributions, **b** modelled median Cloud Condensation Nuclei (CCN) number concentrations as a function of cloud parcel updraft velocity, **c** modelled top of the atmosphere direct aerosol radiative forcing probability distributions caused by new particle formation (NPF) and HOM secondary organic aerosol (HOM SOA) formation, during clear sky conditions, and **d** relative fraction of the modelled CCN number concentrations that are caused by NPF and HOM SOA formation, respectively. The model results were derived based on data from the control run (CTRL), the no NPF simulation (NoNPF), and the no HOM formation simulation (NoHOM). The shaded areas in panels **a**, **b** and **d** show the measured and modelled data range within the 25th to 75th percentiles

Apart from the indirect radiative forcing induced by the altered cloud droplet number concentrations and the optical properties of the clouds, NPF and HOM SOA formation can also influence the Radiative Forcing due to Aerosol–Radiation Interactions (RF$_{ARI}$). In order for particles to scatter light efficiently, they must have a diameter approximately equal to or larger than the wavelength of light. Therefore, mainly particles with diameters >200 nm scatter sunlight back into space and cause a negative RF$_{ARI}$. The RF$_{ARI}$ caused by NPF and HOM SOA formation, during clear sky conditions (no clouds), were estimated using the radiative transfer module implemented in ADCHEM[33,44] (see Methods).

Table 2 summarizes the RF$_{ARI}$ and the relative change in the number of CCN at $w = 0.1$ and $1.0$ m s$^{-1}$ that are a result of NPF, NPF formed exclusively from NH$_3$-H$_2$SO$_4$ clustering, HOM SOA formation, and NPF and HOM SOA formation together. Figure 7a shows the measured and modelled median particle number size distributions at SMEAR II for the CTRL, NoNPF, and NoHOM simulations. The modelled median particle number concentrations from the CTRL simulations are within 25% of the observed particle number concentrations for all particle sizes in the size range from 25 to 400 nm in diameter. The measured and modelled median total particle number concentration in the Aitken-accumulation mode size range (25–1000 nm in diameter) (PN$_{25–1000\,nm}$) are 1727 and 1629 cm$^{-3}$, respectively. Without

NPF PN$_{25–1000\,nm} = 533$ cm$^{-3}$ and without HOM formation PN$_{25–1000\,nm} = 1308$ cm$^{-3}$.

According to our adiabatic cloud parcel model simulations, NPF contributes substantially to the number of CCN at updraft velocities ($w$) >1 m s$^{-1}$ (Fig. 7b, d). Generally, cumulus clouds have updraft velocities >1 m s$^{-1}$, while in stratiform clouds $w$ rarely exceeds 0.5 m s$^{-1}$ refs.[45,46]. At $w \geq 3$ m s$^{-1}$, which results in maximum cloud water vapour supersaturation (SS$_{max}$) $\geq 0.66\%$ (Supplementary Fig. 14), NPF increases the number of CCN by $\geq 50\%$ (Fig. 7d). However, at $w < 0.2$ m s$^{-1}$ (SS$_{max} < 0.25\%$), NPF reduces the number of CCN (Fig. 7d). At an updraft velocity of 0.2 m s$^{-1}$ the median minimum dry diameter of the particles that activate into cloud droplets is 138 nm, both for the CTRL and NoNPF simulations (Supplementary Fig. 14). From Fig. 7a it is apparent that the concentration of particles larger than ~170 nm in diameter is higher in the NoNPF run compared to the CTRL run. The reason for this is that without NPF the condensing material (mainly organics, sulfuric acid, and ammonia) is distributed exclusively onto fewer but substantially larger particles originating from primary particle emissions.

In contrast to NPF, HOM SOA formation increases the CCN concentrations for all updraft velocities. The largest impact is found at updraft velocities $\geq 0.3$ m s$^{-1}$ (SS$_{max} \geq 0.28\%$) where HOM increases the CCN concentrations by 8–12% (Fig. 7d). At $w \leq 0.1$ m s$^{-1}$ (SS$_{max} \leq 0.19\%$) the condensation of HOM

contributes to < 3% of the modelled CCN number concentration. The results from our detailed process-based model simulations are qualitatively consistent with the global model simulations performed by Jokinen et al.[10], which concluded that ELVOC (i.e. HOM according to the definition of that study) have a very minor impact on the CCN at $SS_{max} = 0.2\%$ but clearly increase the number of CCN at $SS_{max} = 1\%$ over the Boreal forest.

HOM contributes to the growth of particles in all size ranges, but mainly increases the particle concentrations in the size range 50–200 nm in dry diameter (Fig. 7a). These relatively small particles also take up other condensable vapours (e.g. semi-VOCs) that otherwise would have partitioned onto larger particles which scatter more sunlight. Thus, it is not unambiguous to estimate how HOM SOA formation influences the direct aerosol–radiation interactions, i.e. if the $RF_{ARI}$ becomes negative or positive. Figure 7c shows the probability distributions of the modelled top of the atmosphere direct aerosol radiative forcing caused by NPF and HOM SOA formation during the periods 15–24 May 2013 and 15 April to 5 May 2014 at SMEAR II, assuming clear sky conditions. The HOM SOA formation generally, but not always, leads to negative $RF_{ARI}$, with an average $RF_{ARI}$ of $-0.10$ W m$^{-2}$. NPF, on the other hand, almost always leads to positive $RF_{ARI}$, with an average value of $+0.15$ W m$^{-2}$. The reason is that NPF indirectly causes a reduction in the number concentration of particles larger than ~170 nm in dry particle diameter (Fig. 7a). We can also conclude, based on the NoNPF-NoHOM simulation, that the negative radiative forcing induced by the HOM SOA formation is offset by the positive $RF_{ARI}$ because of NPF, resulting in average net zero $RF_{ARI}$ for the studied spring periods (see Table 2).

According to our model simulations the NPF at, and upwind from, SMEAR II can largely be explained by sulfuric acid clustering with ammonia. Sulfuric acid is mainly originating from anthropogenic $SO_2$ emissions. Thus, the sulfuric acid induced NPF, which causes an average positive $RF_{ARI}$ of $+0.15$ W m$^{-2}$, reduces the magnitude of the negative $RF_{ARI}$ attributed to the anthropogenic $SO_2$ emissions. Globally the $RF_{ARI}$ of anthropogenic sulfate aerosol particles is estimated to be $-0.4$ ($-0.6$ to $-0.2$) W m$^{-2}$ (ref. [47]).

**Recommendations to the atmospheric modelling community**. Climate and CTMs need to represent the formation and losses of HOM more realistically in order to improve the predictions of SOA formation and its implications for air quality and climate on Earth[1,18,19]. In this work, we have developed and used the near explicit PRAM to provide a complete closure between the modelled and observed HOM concentrations at the Boreal forest station SMEAR II. This fundamental process knowledge is required to efficiently improve the representation of HOM SOA formation in atmospheric models. However, for most large-scale atmospheric model applications, PRAM and the SOA formation scheme used in this work may need to be reduced (simplified). Still, it is important that any reduced mechanism should be able to represent the non-linear SOA yield effects caused by different $O_3$, OH, $RO_2$, $NO_x$, and temperature conditions. If the gas-phase chemistry mechanism (e.g. PRAM) can fulfil these requirements, the particle growth and/or SOA mass formation may be successfully parameterized by lumping the formed closed shell species into a volatility basis set framework[18,23,34]. This can reduce the number of condensable compounds from several hundreds to less than ten. PRAM explicitly treats $RO_2 + RO_2$ reactions between 94 $RO_2$ from the MCMv3.3.1 chemistry and 17 PRAM specific $RO_2$ (Supplementary Table 4), in total 1598 reactions. However, instead of representing the reactions between individual $RO_2$ the dimer formation may be parameterized assuming that

the total concentration of $RO_2$ in MCMv3.3.1 (the so-called $RO_2$ pool)[7] reacts with the $RO_2$ in PRAM using single collective rate coefficients. If the formed HOM dimers are only represented by two dimers, one for the HOM formed from monoterpene ozonolysis and one for dimers formed from OH-oxidation of monoterpenes the total number of reaction in PRAM is reduced from 1773 to 192 and the total number of species from 208 to 89 (Supplementary Table 7). In this case the formed HOM dimers will have to be represented with some average properties of typical HOM dimers, e.g. molar mass and $p_0$. However, since the majority of HOM dimers formed from monoterpenes are ELVOCs, which condenses irreversible to the existing aerosol particles, this simplification can be acceptable from a SOA mass formation perspective. At SMEAR II the reduced PRAM version (Supplementary Table 7) gives almost identical average total HOM concentrations and only 6% higher average HOM dimer concentrations compared to the default PRAM version (Supplementary Table 4) (see Supplementary Fig. 15, Supplementary Table 8). The close agreement between the two PRAM versions and the observations at SMEAR II is partly reflected by that the dominant source of $RO_2$ at SMEAR II are the locally emitted monoterpenes. Thus, in order to conclude about the applicability of the reduced and full PRAM versions for global scale model applications, they should be evaluated also for conditions where a major fraction of the $RO_2$ pool is originating from precursors that do not contribute substantially to HOM formation, e.g., in environments with high isoprene concentrations.

## Discussion

In this work we have developed the first comprehensive PRAM that describes the formation of HOM from monoterpenes. With PRAM implemented, our aerosol dynamics models capture the observed HOM(g) concentrations and SOA mass formation, both during smog chamber experiments and in the atmosphere. During typical spring-time conditions HOM SOA contributes to 18% of the modelled submicron particle mass at the SMEAR II station in Finland.

We show that the combination of NPF and particle growth by biogenic HOM SOA has a profound but complex impact on the aerosol-cloud-climate system over the Boreal forest. In spring the HOM SOA increases the number of CCN with ~10% at cloud updraft velocities ($w$) in the range 0.3–5 m s$^{-1}$, which corresponds to water vapour supersaturations between ~0.28% and ~0.84%. Furthermore, we estimate that the HOM SOA contributes to an average direct aerosol radiative forcing of $-0.10$ W m$^{-2}$. Thus, biogenic HOM SOA formation most likely contributes to climate cooling over the Boreal forest, both with and without the presence of clouds.

The net climate impact (i.e. cooling or warming) caused by NPF over the Boreal forest is more complex, and varies depending on the amount and type of clouds. Our model simulations reveal that the observed NPF can to a large extent, be explained by sulfuric acid clustering with ammonia. In spring the NPF upwind of SMEAR II contributes to ≥50% of the number of CCN at cloud updraft velocities ≥3 m s$^{-1}$, but even reduces the number of CCN at updraft velocities <0.2 m s$^{-1}$. The NPF also causes a positive direct aerosol radiative forcing of on average $+0.15$ W m$^{-2}$ at clear sky conditions. Thus, without clouds or during conditions with stratiform clouds with updraft velocities <0.2 m s$^{-1}$, NPF most likely results in climate warming, while in the presence of cumulus clouds (which typically have $w$ >1 m s$^{-1}$) it will lead to optically thicker clouds and climate cooling. The combined effect of HOM formation and NPF over the Boreal forest is a substantial increase in the number concentration of CCN at cloud updraft velocities >0.2 m s$^{-1}$. However, at clear sky

conditions the negative radiative forcing from HOM SOA formation is offset by the positive radiative forcing caused by the NPF.

We demonstrate that the comprehensive PRAM mechanism may be substantially reduced. The reduced PRAM version can likely be used for realistic representations of HOM SOA formation in regional and global scale CTMs. However, before PRAM is used for large-scale CTM applications we recommend that the mechanism is evaluated also for other regions, e.g., over tropical forests and urban areas.

## Methods

**The PRAM.** We have developed a PRAM[27] for monoterpenes reacting with $O_3$ or OH and coupled it to the Master Chemical Mechanism version 3.3.1 (MCMv3.3.1) using the Kinetic PreProcessor (KPP)[48]. Below we give a general description of the theory behind PRAM. In the Supplementary Information we include tables listing: the molar yield of formation of the initial $RO_2$ that are formed when monoterpenes are oxidized by $O_3$ and OH in PRAM (Supplementary Table 1), all 132 PRAM species formed from ozonolysis of monoterpenes (Supplementary Table 2a), all 76 PRAM species formed from OH oxidation of monoterpenes (Supplementary Table 2b), the MCMv3.3.1 species that are influencing/influenced by the reactions in PRAM (Supplementary Table 3), and the complete PRAM, with all 1773 reactions and reaction rates (Supplementary Table 4).

PRAM simulates the formation of peroxy radical ($RO_2$) autoxidation products (including HOM) formed from ozonolysis and OH-oxidation of monoterpenes. PRAM has been developed based on experimental and theoretical studies of $\alpha$-pinene oxidized by $O_3$ and OH[9–22]. However, for the atmospheric model simulations presented in this work we use PRAM as a general mechanism describing the autoxidation and formation of HOM from four different monoterpenes ($\alpha$-pinene, $\beta$-pinene, limonene and carene), by assigning species specific molar yields of formation of the first $RO_2$ that initiates the autoxidation chain (Supplementary Table 1). These yields provide upper limit estimates of the HOM molar yields for the different monoterpenes+oxidant reactions. After the first reactions that initiate the autoxidation in PRAM, the mechanism does not differentiate between the products formed from the different monoterpenes. In total, PRAM includes 208 species, of which 132 represents the autoxidation products formed from the ozonolysis of monoterpenes and 76 the species formed after the OH-oxidation of monoterpenes. In total, PRAM consists of 1773 reactions listed in Supplementary Table 4.

PRAM explicitly simulates how the autoxidation proceeds via a chain of sequential intramolecular peroxy radical hydrogen shifts (H-shifts) and $O_2$ additions (R1). According to quantum chemical calculations, the activation energies for autoxidation of different $RO_2$ from $\alpha$-pinene are between 22 and 29 kcal mol$^{-1}$ (ref. [14]). This leads to reduced autoxidation rates at low temperatures. PRAM presently uses temperature-dependent autoxidation rates corresponding to an activation energy barrier of 24 kcal mol$^{-1}$ for all autoxidation reactions (Supplementary Table 4, R11–R18 and R1155–1158).

The autoxidation can be terminated by bimolecular reactions where the formed $RO_2$ react with NO, HO$_2$, or other peroxy radicals (R2, R5, R6). When two peroxy radicals react with each other, the product(s) will either be alkoxy radicals (RO) (R6a), closed-shell monomers (R6b, c) or a dimer (R6d).

$$C_{10}H_{15}O_2 \cdot \xrightarrow{H-shift+O_2} C_{10}H_{15}O_4O_2 \cdot \xrightarrow{H-shift+O_2} C_{10}H_{15}O_6O_2 \cdot \xrightarrow{H-shift+O_2} C_{10}H_{15}O_8O_2 \cdot \ldots \quad (R1)$$

$$C_{10}H_{15}O_xO_2 \cdot + NO \rightarrow C_{10}H_{15}O_xNO_3 \quad (R2a)$$

$$\rightarrow C_{10}H_{15}O_xO \cdot + NO_2 \quad (R2b)$$

$$C_{10}H_{15}O_xO \cdot + O_2 \rightarrow C_{10}H_{15}O_{x+1}O_2 \cdot \quad (R3a)$$

$$\rightarrow C_{10}H_{14}O_xO + HO_2 \quad (R3b)$$

$$C_{10}H_{15}O_xO \cdot \rightarrow \text{fragmentation products} \quad (R4)$$

$$C_{10}H_{15}O_xO_2 \cdot + HO_2 \rightarrow C_{10}H_{15}O_xOOH + O_2 \quad (R5)$$

$$C_{10}H_{15}O_xO_2 \cdot + RO_2 \cdot \rightarrow C_{10}H_{15}O_xO \cdot + RO \cdot + O_2 \quad (R6a)$$

$$\rightarrow C_{10}H_{14}O_xO + ROH + O_2 \quad (R6b)$$

$$\rightarrow C_{10}H_{15}O_xOH + RCHO + O_2 \quad (R6c)$$

$$\rightarrow C_{20}H_{30}O_{x+2} + O_2 \quad (R6d)$$

PRAM uses the same reaction rate coefficients for all $RO_2$ + NO and $RO_2$ + HO$_2$ reactions, with values identical to MCMv3.3.1 (Supplementary Table 4). For the

$RO_2$ + HO$_2$ reactions the formed product is always a closed-shell molecule with a hydroperoxide functional group (–OOH) replacing the peroxy radical group (–OO·) (R5). For the $RO_2$ + NO reactions the branching ratio between the channel forming organonitrate HOM (R2a) and the channel forming an alkoxy radical and NO$_2$ (R2b) varies depending on the $RO_2$ species. The RO formed from (R2b and R6a) is either rapidly isomerizing to a hydroxyl-substituted alkyl radical that further react with $O_2$ and form a new $RO_2$ species (R3a), form a closed-shell HOM monomer with an additional carbonyl group (R3b), or decompose and form more volatile species (R4)[49]. In the mechanism the fragmentation products from R4 are represented by the MCM species C717O2 (an $RO_2$) and CH3COCH3 (acetone). PRAM assumes that all $RO_2$ + $RO_2$ reactions (R6) exclusively occur between $RO_2$ formed from the PRAM autoxidation mechanism (R1) and in total 94 $RO_2$ species formed in the MCMv3.3.1 chemical mechanism. These are all $RO_2$ with 10, 9, or 8 carbon atoms originating from oxidation of $\alpha$-pinene, $\beta$-pinene, and limonene in MCM.

When $\alpha$-pinene is oxidized by $O_3$ one of the two ring structures is broken but a cyclobutyl ring is left intact in the $RO_2$ isomers ($C_{10}H_{15}O_4$) that are formed. According to quantum chemical calculations by Kurtén et al.[22] the cyclobutyl ring inhibits multiple autoxidation steps and prevents the first intramolecular H-shifts reactions rates to exceed 0.3 s$^{-1}$ at 298 K. In PRAM we therefore assigned a rate constants of 0.3 s$^{-1}$ at 298 K for the first H-shift reaction (R11 in Supplementary Table 4). Kurtén et al. also examined possible reaction pathways that can lead to opening of the cyclobutyl ring. According to Kurtén et al. the ring opening can likely occur via alkoxy radicals. Such pathways are also present in MCMv3.3.1 (refs. [7,28,29]) when the $RO_2$ isomer, with the MCM name C107O2, react with NO, NO$_3$ or other $RO_2$ and form an alkoxy radical C107O that can isomerize and react with $O_2$ and form a $C_{10}H_{15}O_5$ peroxy radical named C108O2. In PRAM we therefore included the possibility of such additional HOM formation pathway for $\alpha$-pinene, which is initiated by the reaction between C107O2 and other $RO_2$ (R1152 followed by R20 and R12–R19 in Supplementary Table 4). In the Jülich Plant Atmosphere Chamber (JPAC) model simulations (Fig. 1), this additional HOM formation channel accounts for ~30% of the formed HOM. However, the actual contribution of this potential HOM formation pathway compared to the autoxidation pathway via the $C_{10}H_{15}O_4$ isomers cannot be validated based on the experiments. The idea of having at least two different HOM formation pathways from $\alpha$-pinene ozonolysis, one which lead to rapid HOM formation (on the order of 10 s) and a second slower pathway, e.g. that requires alkoxy radical formation via bimolecular reactions, is supported by the fact that the HOM yields reported from smog chamber experiments with 45 min reaction time[9] are a factor of ~2 higher than the yields from flow tube experiments with 40 s reaction time[10].

**PRAM sensitivity analysis.** In MCMv3.3.1 the generic reaction rate constants for $RO_2$ + $RO_2$ reactions range from $10^{-11}$ cm$^3$ molecule$^{-1}$ s$^{-1}$ for acyl peroxy radicals to $6.7 \times 10^{-15}$ cm$^3$ molecule$^{-1}$ s$^{-1}$ for tertiary carbon peroxy radicals[7,28,29]. These rate coefficients are based on measured reaction rates of generally less oxygenated and smaller $RO_2$ molecules than the $RO_2$ in PRAM. PRAM uses $RO_2$ + $RO_2$ reaction rates leading to closed shell monomers in the range of $5 \times 10^{-12}$ cm$^3$ molecule$^{-1}$ s$^{-1}$ to $10^{-11}$ cm$^3$ molecule$^{-1}$ s$^{-1}$ and $RO_2$ + $RO_2$ reaction rates leading to closed shell HOM dimers between $10^{-13}$ cm$^3$ molecule$^{-1}$ s$^{-1}$ to $5 \times 10^{-12}$ cm$^3$ molecule$^{-1}$ s$^{-1}$, with the lowest values applied to the reactions involving the least oxygenated peroxy radicals and the highest values for the most oxygenated peroxy radicals (Supplementary Table 4). With the reaction rate coefficients as tabulated in Supplementary Table 4, PRAM match the observed HOM $RO_2$ concentrations in JPAC (Fig. 1c) and the trends and absolute concentrations of HOM $RO_2$ in the atmosphere reasonably well (Fig. 3a). Supplementary Figure 16 compares the observed and modelled HOM concentrations in JPAC when we scale all $RO_2$ + $RO_2$ reaction rates ($k(RO_2 + RO_2)$) in PRAM up or down with a factor of two. With $k(RO_2 + RO_2) \times 0.5$, the lifetime and concentration of $RO_2$ increases. At an atmospheric relevant $\alpha$-pinene + $O_3$ reaction rate of 0.3 ppt$_v$ s$^{-1}$ and $k(RO_2 + RO_2) \times 0.5$ the HOM $RO_2$ concentrations become 60% higher, while with $k(RO_2 + RO_2) \times 2$ the modelled HOM $RO_2$ become 40% lower. At low $\alpha$-pinene + $O_3$ reaction rates the modelled closed shell HOM formation is limited by the formation of highly oxygenated $RO_2$ via autoxidation and by the bimolecular termination reactions that lead to closed shell products. At conditions with high $\alpha$-pinene + $O_3$ reaction rates (high absolute $RO_2$ concentrations) the closed shell HOM formation are primarily limited by the formation of the highly oxygenated $RO_2$ and not by the bimolecular termination reactions. Instead, $RO_2$ + $RO_2$ reactions can cause termination of the autoxidation reaction chain before many of the $RO_2$ become highly oxygenated. Thus, in the low $RO_2$ concentration regime, higher $k(RO_2 + RO_2)$ results in higher closed shell HOM concentrations, while in the high $RO_2$ concentration regime, higher $k(RO_2 + RO_2)$ results in lower closed shell HOM concentrations. This is why the modelled closed shell HOM concentrations (monomers and dimers) are slightly higher at $\alpha$-pinene + $O_3$ reaction rates <0.4 ppt$_v$ s$^{-1}$ but lower at $\alpha$-pinene + $O_3$ reaction rates >0.4 ppt$_v$ s$^{-1}$ in the $k(RO_2 + RO_2) \times 2$ run compared to default PRAM setup ($k(RO_2 + RO_2) \times 1$). The opposite trends can be seen for the $k(RO_2 + RO_2) \times 0.5$ simulation. The HOM observations in JPAC indicate that the absolute closed shell HOM monomer and dimer yields decreases somewhat when the $\alpha$-pinene + $O_3$ reaction rates increases in the chamber. These results are consistent with the PRAM model simulations which uses $k(RO_2 + RO_2) \times 2$. However, at atmospheric

relevant α-pinene + $O_3$ reaction rates <0.5 $ppt_v$ $s^{-1}$ the modelled closed shell HOM concentrations are relatively insensitive to the exact values of $k(RO_2 + RO_2)$ (Supplementary Fig. 16).

The temperature dependence of the autoxidation reaction rates in PRAM (Supplementary Table 4) all corresponds to an activation energy of 100 kJ for the rate limiting H-shifts ($E_{H\text{-shift}}$). This activation energy is within the range of values suggested by Rissanen et al.[14], which calculated $E_{H\text{-shift}}$ in the range of 90–120 kJ for different $RO_2$ isomers formed from ozonolysis of α-pinene. The H-shift activation energy used as default in PRAM is higher than the $E_{H\text{-shift}}$ measured and calculated for peroxy radicals originating from several other VOCs, which generally are in the range 40–80 kJ[11,50,51]. However, Quéléver et al.[52] recently showed that the observed HOM yields during α-pinene ozonolysis experiments in the AURA chamber was about 50 times lower at 273 K compared to 293 K. The AURA experiments were performed using an initial α-pinene and $O_3$ concentrations of 50 and 100 ppb, respectively. This corresponds to an α-pinene + $O_3$ reaction rate of ~10 $ppt_v$ $s^{-1}$. The results from Quéléver et al. indicate that the autoxidation reaction rates of $RO_2$ formed from α-pinene ozonolysis must slow down considerably between 293 and 273 K. This, together with the presumably high absolute $RO_2$ in the AURA experiments (i.e. short lifetime of $RO_2$ with respect to $RO_2 + RO_2$ reactions) may at least partly explain the observed drastic drop in the HOM yield between 293 and 273 K[52]. With the default $E_{H\text{-shift}}$ of 100 kJ in PRAM, the modelled HOM molar yield at an α-pinene + $O_3$ reaction rate of ~1 $ppt_v$ $s^{-1}$ increases from 2.3% to 9.0% between 270 and 310 K, while with $E_{H\text{-shift}} = 50$ kJ the yield range between 4.4% and 8.3% and with $E_{H\text{-shift}} = 150$ kJ the HOM molar yields range between 1.6% and 9.0% (Supplementary Fig. 17). For all model sensitivity tests the absolute autoxidation reaction rates were kept identical at the reference temperature 289 K, i.e. the same temperature as was used during the JPAC experiments[9].

In Supplementary Fig. 15 and Supplementary Table 8 we compare the modelled HOM concentrations with the observations at SMEAR II for different model sensitivity tests where we scaled all $RO_2 + RO_2$ reaction rates in PRAM up or down with a factor of two, or change the activation energy of the H-shift reaction rates from the default 100 to 50 kJ or 150 kJ respectively. The differences in the modelled total HOM concentrations between the different model simulations are relatively small (e.g. FAC2 values between 0.92 and 0.94). Supplementary Fig. 18 compares the average vertical HOM concentration profiles at SMEAR II from the different PRAM sensitivity tests. All concentration profiles are within ±10% from the default PRAM model simulation results at all altitudes. Thus, the modelled HOM concentrations for the simulated period at SMEAR II seem to be relatively robust, considering the estimated range of uncertainty in the absolute $RO_2 + RO_2$ reaction rates and the H-shift activation energies. The small differences between the model sensitivity tests with different values of $E_{H\text{-shift}}$ is related to that the average surface temperatures at SMEAR II were 287.9 K for the simulated period.

**Reduced PRAM version.** In order to be able to implement PRAM into large-scale CTMs the number of reactions and species need to be minimized. Instead of considering reactions between individual $RO_2$ that form HOM dimers (R85–R1118 and R1193–R1756, in Supplementary Table 4), it may adequate and necessary to be represent these reactions using a simplified approach where the total pool of $RO_2$ are allowed to react with the individual $RO_2$ in PRAM using single collective rate coefficients (Supplementary Table 7, R85–R95 and R170–R175). This, drastically reduces the total number of reaction in PRAM from 1773 to 192 and the number of species from 208 to 89. In the JPAC experiments, where all $RO_2$ are originating from α-pinene, this simplification introduces no noticeable model deviation concerning the total HOM gas-phase concentrations or the modelled SOA formation.

**ADCHAM and ADCHEM model setup and description.** In this work we use the Aerosol Dynamics gas- and particle phase chemistry model for laboratory CHAMber studies (ADCHAM[32]) and the trajectory model for Aerosol Dynamics, gas and particle phase CHEMistry and radiative transfer (ADCHEM[33,34]). ADCHAM and ADCHEM use identical aerosol dynamics and gas-phase chemistry codes. They take into account Brownian coagulation and the condensation, dissolution, and evaporation of $H_2SO_4$, $NH_3$, $HNO_3$, and all organic oxidation products from Master Chemical Mechanism version 3.3.1 (MCM v.3.3.1) and from the PRAM mechanism (Supplementary Table 2) with pure liquid saturation vapour pressures ($p_0$) less than $10^{-2}$ Pa (in total, 828 species at 290 K). $p_0$ were estimated with the functional group contribution method SIMPOL.[24]. For the peroxy radical autoxidation products the molecule properties provided in Supplementary Table 2 were used when calculating their $p_0$.

**Smog chamber simulations.** The first-order wall losses of HOM and other organic vapours ($k_{wall,i}$) for the JPAC smog chamber simulations were estimated based the experimentally derived HOM wall losses of HOM monomers[9]. HOM molecules with a molecular formula $C_{10}H_{16}O_8$ have a first-order wall loss rate of 1/75 $s^{-1}$ in JPAC[9]. In the model the individual HOM wall loss rates were estimated by multiplying the experimentally derived wall loss rate for $C_{10}H_{16}O_8$ with the ratio between the molecular diffusion coefficient for $C_{10}H_{16}O_8$ and the molecular diffusion coefficients for the other molecules ($D_i$) (Eq. 1). $D_i$ were calculated based on the Fuller's method[53]. For a dimer with molecular formula $C_{20}H_{30}O_{11}$ the

estimated first-order wall loss rate become 1/100 $s^{-1}$, which is slightly lower than the experimentally derived values for HOM dimers with the same molecular formula of 1/90 $s^{-1}$.

$$k_{wall,i} = k_{wall,C_{10}H_{16}O_8} \frac{D_i}{D_{C_{10}H_{16}O_8}}. \tag{1}$$

All closed-shell HOM species and the HOM peroxy radicals were assumed to deposit irreversibly on the JPAC chamber walls, motivated by their generally low volatility[9] and high reactivity[54,55]. However, for the other condensable organic compounds from MCMv3.3.1 we expect that the VOC wall partitioning are more of a reversible nature. For smog chambers with Teflon walls, model have been developed that takes into account the reversible partitioning of VOC assuming that the Teflon wall itself behaves like a large effective organic mass ($C_{wall}$)[32,56]. For JPAC where the walls are made out of glass, the release of semi-volatile non-reactive VOC back to the gas-phase from the walls ($k_{wall,back,i}$) is most likely higher than for smog chamber made out of Teflon. In this work we still use the theory developed for Teflon walls (Eq. 2) but assume that the $C_{wall}$ is smaller than what generally is used for Teflon smog chamber walls. With a $C_{wall}$ of 5 μmol $m^{-3}$ ADCHAM is able to reproduce the observed SOA mass formation in JPAC (Fig. 2). This value can be compared with literature values of 9, 20, 50 and 120 μmol $m^{-3}$ for alkanes, alkenes, alcohols and ketones absorbing on Teflon walls[56].

$$k_{wall,back,i} = k_{wall,i} \frac{p_{0,i}}{RTC_{wall}}. \tag{2}$$

$p_{0,i}$ in Eq. (2) is the pure liquid saturation vapour pressure of compound $i$, $R$ is the universal gas constant 8.3145 J $mol^{-1}$ $K^{-1}$ and $T$ is the temperature in K.

**Atmospheric model simulations.** In the present study we used ADCHEM as a one-dimensional column model consisting of 40 vertical layers, logarithmically spaced, with intervals increasing from 3 m at the surface to 100 m at the top of the model domain that extends up to 2500 m a. g. l. The main model time step was 30 s. The atmospheric diffusion equation is solved in the vertical direction using diffusion coefficients calculated with a slightly modified Grisogono scheme[57]. The aerosol dynamics include NPF, Brownian coagulation, condensation/evaporation, and dry deposition of particles and gases (including HOM). In and below cloud scavenging of aerosol particles and in-cloud sulfate aerosol formation and scavenging of $SO_2$, $H_2O_2$, $NH_3$, $HNO_3$ and HCHO were also considered, analogous to Roldin et al.[33]. However, the model did not consider any cloud droplet aqueous phase chemistry or in- and below cloud scavenging of other organic molecules. The particle number size distributions are represented with 100 fixed size bins between 1.07 nm and 2.5 μm in dry diameter.

**New particle formation.** The NPF via $NH_3$-$H_2SO_4$ was modelled using the ACDC[38], which solves the time evolution of a population of molecular clusters considering all possible collision and evaporation processes between the clusters and vapour molecules (in this case $NH_3$ and $H_2SO_4$), as well as ionization and recombination by primary gas-phase ions, and cluster scavenging by larger aerosol particles. The NPF rate is obtained as the flux of stable clusters growing out of the size range simulated by ACDC.

ACDC was implemented as an explicit molecular cluster dynamics module, in which the time-dependent cluster concentrations are monitored and updated at every time step. The ACDC module was combined to ADCHEM via an interface that takes as input the concentrations of $H_2SO_4$ and $NH_3$ vapours, the temperature, and the cluster scavenging rate obtained from the particle distribution simulated within ADCHEM. When the module is called, the ambient conditions are updated, and the time evolution of the cluster distribution is solved for the given time step. As output, the module gives the number of >1.07 nm particles that grew out of the ACDC size range during the simulated time interval. These newly-formed nanoparticles are assumed to behave like aerosol particles and are introduced into ADCHEM, which simulates the consecutive condensation growth, evaporation, and losses by coagulation and deposition.

The simulated clusters consist of up to 5 $H_2SO_4$ and 5 $NH_3$ molecules, with an approximate diameter of 1.07 nm. The rate constants related to cluster growth and evaporation were calculated as described by Olenius et al.[38], using previously published quantum chemical data, computed at the B3LYP/CBSB7//RICC2/aug-cc-pV(T+d)Z level of theory, to calculate the cluster evaporation rates. Both electrically neutral and negatively and positively charged clusters were included, with ionization and recombination of molecules and clusters occurring through collisions with generic gas-phase ions $O_2^-$ and $H_3O^{+}$[38]. The production rate of the generic ions was calculated as the sum of the ionization rate due to galactic cosmic rays and ionization caused by the decay of radon. The ionization rate due galactic cosmic was set to 1.7 $cm^{-3}$ $s^{-1}$[30]. The radon concentration and radon induced ionization was modelled using the radon emission map and mean ionization rate formula from Zang et al.[58]. In average the modelled ion production rate was 3 $cm^{-3}$ $s^{-1}$ in the surface layer at the SMEAR II station. Supplementary Figure 19 in the Supplementary Information shows the modelled ion production rate at SMEAR II for the periods 15–24 May 2013 and 15 April–5 May 2014. The size-dependent cluster scavenging rate was calculated based on the condensation sink (CS) of $H_2SO_4$ vapour, given by ADCHEM, according to the power-law formula by

Lehtinen et al.[59]. In this formula, the parameter $m$ was set to $-1.6$, corresponding to typical boundary layer conditions.

NPF via organics-$H_2SO_4$ cluster formation was considered in ADCHEM using a semi-empirical parameterization (Eq. 3), which was developed based on JPAC experiments on real plant emissions[39] and used in our previous ADCHEM model simulations of NPF events at the Pallas field station in Northern Finland[34]. The formation rate of 1.5 nm particles is parameterized as

$$J_{1.5} = k_{SA-Org}(T)[H_2SO_4][ELVOC_{nucl}] \quad (3)$$

$ELVOC_{nucl}$ in Eq. (3) is treated as an effectively non-volatile organic molecule formed as first-generation oxidation product with a molar yield (i.e. probability) of $10^{-5}$ for each monoterpene molecule that is oxidized by OH[34,39]. Similar to Yu et al.[60] the temperature dependence of the pre-factor $k_{SA-Org}$ in Eq. (1) was estimated using the calculated Gibbs free energy at 298 K ($\Delta G$) and entropy change ($\Delta S$) for the critical cluster formation (Eqs. 4 and 5). $\Delta G$ was set to $-15.1$ kcal mol$^{-1}$ and $\Delta S$ to $-61.1$ cal mol$^{-1}$ K$^{-1}$ according to the quantum chemical calculations performed by Elm et al.[61] These values correspond to the clustering of a sulfuric acid molecule with an idealized case of a large dicarboxylic acid molecule with weak intramolecular hydrogen bonds and two direct sulfuric acid–carboxylic acid interactions.

$$k_{SA-Org} = 5 \times 10^{-13} e^{\left(\frac{-\Delta H}{R}\left(\frac{1}{T} - \frac{1}{T_0}\right)\right)} \quad (4)$$

$$\Delta H = \Delta G - \Delta S \cdot T \quad (5)$$

$\Delta H$ in Eqs. (4) and (5) is the enthalpy change and $T_0$ is a reference temperature set to 298 K. In our previous applications of ADCHEM for simulations of NPF and growth events during the summertime in Northern Finland[34] we used a temperature-independent NPF pre-factor of $2 \times 10^{-11}$ cm$^3$ s$^{-1}$. With Eq. (4) $k_{SA-Org}$ reach this value at a temperature of 279 K, which corresponds well with typical summertime temperatures in the boundary layer in Northern Finland.

**Gas and primary particle emissions**. The emissions of $\alpha$-pinene, carene, $\beta$-pinene, and limonene were modelled with a 1D version of MEGAN 2.04 (Model of Emissions of Gases and Aerosols from Nature)[40,41]. The individual monoterpene emissions were estimated based on the measurements from 40 Scots pine trees around SMEAR II, which on average emit 43.7% $\alpha$-pinene, 39.6% carene, 9% $\beta$-pinene, and 2.3% limonene[62]. In ADCHEM the three lowermost model layers at 0–3, 3–9, and 9–18 m are within the forest canopy at SMEAR II, and MEGAN simulates the BVOC emissions in each of these layers.

Of all monoterpenes MCMv3.3.1 only includes chemical mechanisms for $\alpha$-pinene, $\beta$-pinene, and limonene. However, since almost 40% of the monoterpenes emitted at SMEAR II is carene we include the initial oxidation reactions between carene and OH, $O_3$ or $NO_3$ in the gas-phase chemistry mechanism, and assume that the formed oxidation products are identical to the products formed from the $\alpha$-pinene oxidation in MCMv3.3.1. This also means that we assume that carene forms HOM with the same yield and composition as $\alpha$-pinene. Like $\alpha$-pinene, carene has one endocyclic double bond and is therefore expected to have high HOM formation yields. However, future research is needed to obtain more detailed chemical information on carene processes.

Gas-phase emission of dimethyl sulfide (DMS) in the marine boundary layer was estimated based on monthly mean seawater concentrations[63] and a sea-to-air transfer velocity parameterization[64]. Anthropogenic emissions of $NO_x$, $SO_2$, CO, $NH_3$, VOC with a resolution of $0.1° \times 0.1°$ were retrieved from EMEP (European Monitoring and Evaluation Programme) database (EMEP/CEIP 2014, present state of emissions as used in EMEP models)[65]. Supplementary Table 9 lists all VOC species that were used to represent the anthropogenic and biogenic VOC in ADCHEM. Size-resolved anthropogenic continental primary particle emissions were derived from a global $0.5° \times 0.5°$ emission inventory[66]. Primary particle emission from ship traffic was parameterized based on the gas-phase emission of $SO_2$ by using a conversion factor of $8.33 \times 10^{14}$ particles/(g $SO_2$)[67]. The size distribution of the primary particles from ships was estimated based on a study by Jonsson et al.[68]. Primary particle emissions of wind-generated marine aerosols were also included[69]. The model was initialized with an aerosol particle number concentration of 100 cm$^{-3}$ in each vertical layer, with a unimodal lognormal particle number size distribution having a geometric mean diameter of 120 nm and a geometric standard deviation of 2. Ninety per cent and 10% of the dry particle volume in each size bin was assumed to be composed of non-volatile organic material and AS respectively.

**Adiabatic cloud parcel model simulations**. We used an adiabatic cloud parcel model[42,43] to calculate the number of activated cloud droplets and different predefine updraft velocities at SMEAR II. For the calculations we assume that all organic compounds (including HOM) are fully water soluble at the point of cloud droplet activation. As input we used the modelled aerosol particle properties in the surface layer from the CTRL, NoNPF and NoHOM runs from the periods 15–24 May 2013 and 15 April to 5 May 2014. The air parcels were assumed to start at the surface (0 m) with an initial relative humidity (RH) of 95% and rise to an altitude of 500 m with fixed predefined updraft velocities of 0.01, 0.02, 0.05, 0.1, 0.2, 0.3, 0.4, 0.5, 0.6, 0.7, 0.8, 0.9, 1.0, 1.5, 2.0, 3.0, 5.0 or 10 m s$^{-1}$. With this setup the air gets

supersaturated (RH > 100%) with respect to water vapour at ~115 m altitude and a fraction of the particles (the CCN) gets activated into cloud droplets. Depending on the updraft velocity the maximum cloud supersaturation ($SS_{max}$) is reached within a few metres to a few tens of metres above the cloud base (see Supplementary Fig. 13).

**Aerosol radiative forcing calculations**. In order to estimate the direct aerosol radiative forcing at SMEAR II, caused by the NPF and HOM SOA formation upwind the station, we used the radiative transfer scheme implemented into ADCHEM[33]. This scheme is based on the quadrature two-stream approximation scheme from Toon et al.[44]. We have used this scheme previously to estimate the radiative forcing caused by anthropogenic aerosol particle emissions in urban plumes[42]. In this work we calculated the top of the atmosphere net downward shortwave radiation at SMEAR II for the periods 15–24 May 2013 and 15 April to 5 May 2014 using the modelled vertical column aerosol particle properties, including the modelled aerosol particle liquid water content from the CTRL and the different sensitivity model runs. Further we assumed clear sky conditions with no aerosol particles above 2500 m altitude, an ozone column depth of 300 Dobson units and a surface albedo of 0.11 (ref. [70]). The $RF_{ARI}$ caused by NPF and HOM SOA formation were estimated by calculating the difference in the modelled top of the atmosphere net downward shortwave radiation between the CTRL run and the NoNPF and NoHOM runs respectively.

**Simulation period and location**. ADCHEM was implemented for simulations of aerosol particle and trace gas concentrations during 10 days in 2013 (15–24 May) and 21 days in 2014 (15 April to 5 May) at the SMEAR II field station in Finland (61.85°N, 24.28°E). For the 2013 period, ADCHEM was first operated as a stationary column model at SMEAR II, with the purpose to evaluate the PRAM mechanism. ADCHEM was continuously reading in the observed concentrations of $NO_2$, $O_3$, CO and total monoterpene at eight different altitudes between 4.2 and 125 m above ground level (a.g.l.), $SO_2$ at 16.8 m and particle number size distributions for particles between 2.8 and 1000 nm in diameter. The particle number size distribution was assumed to be constant in the whole model domain (0–2500 m a.g.l). The modelled total monoterpene concentrations within the lowermost 125 m were scaled for each model time step in order to match the observed total monoterpene concentrations at SMEAR II. All other trace gas concentrations as well as the particle chemical composition and hygroscopic growth were calculated in order to provide the most realistic condensation losses of HOM and other condensable compounds.

During spring 2013 HOM were measured at ~2 m altitude using a nitrate-ion-based chemical ionization atmospheric pressure-interface time-of-flight mass spectrometer (CI-APi-TOF)[31]. Unfortunately, only a few NPF events could be observed at SMEAR II during spring 2013. Thus, in order to be able to evaluate the HOM contribution to the growth of new particles of ~1 nm in diameter into the CCN size range, we also applied ADCHEM for a second period in spring 2014. During this period, NPF events with consecutive growth could be observed at more than 50% of the days (Fig. 5a). Both for the spring 2013 and the spring 2014 periods ADCHEM was operated as a Lagrangian vertical column model following in total 248 individual air mass trajectories starting 4 days backward in time before they reached SMEAR II, with 3 h between the arrival of each trajectory, covering in total 10 consecutive days during 2013 and 21 consecutive days during 2014. The trajectories were calculated with the Hybrid Single Particle Lagrangian Integrated Trajectory Model (HYSPLIT) with meteorological data from the Global Data Assimilation System (GDAS), downloaded from NOAA Air Resource Laboratory Real-time Environmental Application and Display sYstem (READY)[71].

**Henry's law coefficients of HOM**. In order to be able to calculate the dry deposition losses of HOM with the dry deposition resistance model in ADCHEM[72], the Henry's law coefficients ($H_i$) of the HOM species were estimated based on their pure liquid saturation vapour pressures ($p_{0,i}$) and their activity coefficients in an infinitesimally dilute aqueous solution (Eq. 6). In ADCHEM the Henry´s law coefficients are used when calculating the resistance to uptake at the surface. However, if $H$ is larger than ~$10^8$ M atm$^{-1}$ the surface resistance become negligible compared to the aerodynamic- and quasilaminar resistances and the dry deposition is only limited by the atmospheric turbulence close to the surfaces. The activity coefficients ($\gamma_i$) were derived with the AIMOFAC model[73], which is implemented into ADCHAM[32]. $M_{H_2O}$ and $\rho_{H_2O}$ in Eq. (6) are the molar mass and density of water respectively.

$$H_i = \frac{\rho_{H_2O}}{p_{0,i} \cdot M_{H_2O} \cdot \gamma_i} \quad (6)$$

For three of the least oxidized and most volatile HOM species in PRAM (C10H16O6iso1, C10H16O6iso2 and C10H18O6 in Supplementary Table 2 and Supplementary Fig. 20) we compared the $H_i$ from SIMPOL-AIOMFAC with the $H_i$ computed using the BP-TZVPD-FINE parameterization in COSMOTherm[74] (Supplementary Table 6). Briefly, this involves relatively inexpensive quantum chemical density functional calculations on the molecules of interest (in this case the three HOM species, as well as water) both in vacuum and using COSMO-RS[75], which is a type of continuum solvent model (CSM)[76]. The COSMO-calculation

yields input files describing the screening charge surface (known as the σ-surface) of the molecule, which is converted into a distribution function (known as the σ-profile). In somewhat simplified terms, the σ-profile corresponds to the relative amount of the molecule's surface with a certain polarity. The intermolecular electrostatic, hydrogen bonding, and van der Waals interactions between surface segments with different σ values (polarities) can then be computed within the COSMOTherm programme using various parameterizations, of which BP-TZVPD-FINE is the most advanced (and computationally expensive, especially for systems with multiple H-bonding groups). Once the intra- and intermolecular interactions are thus described, the chemical potential of the HOM molecules in a water solution, and finally the Henry's law coefficients, can be computed. Molecular conformations were generated using the systematic conformational search algorithm of the Spartan 14 program[77] and the MMFF force-field. This yielded a few tens of conformers for C10H16O6iso2 and C10H16O6iso3, and about 1300 conformers for C10H16O6iso1. COSMO input files were then generated for all conformers of all three structures. (The input files for H2O are available in the COSMOTherm library.) The COSMOconf program (part of the same programme suite as COSMOTherm) was subsequently used to detect unique conformers (i.e. eliminate possible duplicate structures and structures with similar chemical potentials in a pre-defined set of mixtures from the initial set of COSMO input files), and to map the geometries of the conformers in the gas phase and in the solvent to ensure that each conformer in the solvent has a corresponding conformer in the gas phase. For the C10H16O6iso1 case, the 100 lowest-energy conformers were selected (as this turned out to be the largest number of conformers that was computationally feasible to treat with the current version of COSMOTherm).

For C10H16O6iso2 and C10H18O6 the difference between the H from the two methods is less than one order of magnitude. However, for C10H16O6iso1 H is more than two orders of magnitude larger according to SIMPOL-AIMOFAC compared to the value derived with COSMOTherm using all 100 conformers. The large discrepancy between COSMOTherm and SIMPOL+AIMOFAC predictions for C10H16O6iso1 may be related to an overestimation of the strength of the intramolecular H-bonds of the molecule (relative to the intermolecular H-bonds with water) by COSMOTherm. A similar effect has been noted concerning HOM vapour pressures[6,78]. The two other molecules could be less affected by this due to steric constraints of the ring structures preventing or at least weakening some of the intramolecular H-bonding interactions. We tested this hypothesis by computing H values for all three species using only conformers with a minimal number of H-bonds (which turned out to be 0 full and 0 partial H-bonds for C10H16O6iso1 and C10H16O6iso2, and 0 full and 1 partial H-bond for C10H18O6). This approach was recently observed to improve agreement with experiments for saturation vapour pressures of two HOM-like isoprene oxidation products[78]. Using only conformers with a minimal number of H-bonds led to a significant increase in H for all three species, with the largest increases (around a factor of 300) surprisingly observed for C10H16O6iso2 and C10H18O6. This indicates that contrary to our hypothesis, intramolecular H-bonding interactions are, according to COSMOTherm, even stronger in these species than in C10H16O6iso1. As expected, the approach improved agreement with SIMPOL+AIMOFAC for C10H16O6iso1, though the agreement decreased for the other two cases. After correcting for the potential overestimation of intramolecular H-bonding, the COSMOTherm−predicted H values are well above $10^9$ M atm$^{-1}$ for all three studied species. Therefore, the approximately two order of magnitude uncertainty in the calculated H (indicated by both the difference between COSMOTherm and SIMPOL+AIMOFAC values, and the variation of COSMOTherm values depending on number of intramolecular H-bonds allowed in the conformers) is not crucial for the dry deposition losses of C10H16O6iso1 and most other peroxy radical autoxidation products listed in Supplementary Table 6.

**Mass spectrometer measurements at the SMEAR II station**. The HOM concentrations in the JPAC chamber were measured with a nitrate-ion-based chemical ionization atmospheric pressure-interface time-of-flight mass spectrometer (CI-APi-TOF)[9]. An identical CI-APi-TOF, described by Yan et al.[31] were also used at SMEAR II during spring 2013 to measure the HOM and H2SO4 concentrations at ~2 m above ground level. The measured organonitrate concentration was represented by the sum of the species $C_{10}H_{15}O_{5-12}NO_3$ and the measured RO2 concentration was represented by the sum of the species $C_{10}H_{15}O_{6-13}$, $C_9H_{13}O_8$, and $C_9H_{13}O_{10}$. We decided to not consider the observed organonitrates with six or seven oxygen atoms because a fraction of them may not be formed via peroxy radical autoxidation. For example, in MCM the compounds NC101O2 and NC102O2 with molecular formulas $C_{10}H_{15}NO_6$ and $C_{10}H_{15}NO_7$ are formed when α-pinene is reacting with NO3 followed by a second oxidation step involving OH. During the α-pinene + NOx ozonolysis experiments in JPAC, with low levels of NO3 radicals, $C_{10}H_{15}NO_6$ and $C_{10}H_{15}NO_7$ accounted for only ~3% of the total observed organonitrate concentrations in the gas-phase. However at SMEAR II they on average account for 26% of the observed total organonitrate concentrations. This illustrates that there are other mechanism than peroxy radical autoxidation that can lead to $C_{10}H_{15}NO_6$ and $C_{10}H_{15}NO_7$. Therefore we decided to only compare the modelled and measured HOM-NO3 with eight or more oxygen

atoms, which most likely almost exclusively are formed via peroxy radical autoxidation.

The measured HOM monomers were calculated as all the compounds with even mass in the mass range of 290–450 u (mass of the reagent ion NO3$^-$ included) and the HOM dimers are all compounds with even mass in the range 452–590 u, except for some contamination peaks that were subtracted from the sum.

During spring 2013 the particle chemical composition at SMEAR II was measured with an ACSM[79] while during the spring 2014 we used data from an Aerodyne high resolution time-of-flight AMS[80]. In summary, AMS and ACSM are capable of measuring the non-refractory composition of submicrometre aerosol particles by applying thermal vapourization and electron impact (EI) ionization. General principles of AMS measurements, calibrations, modes of operation and data processing have been described comprehensively in previous work[81,82].

The AMS data were processed using Tof-AMS Data Analysis Toolkit SQUIRREL version 1.57H and PIKA version 1.16H in Igor Pro software (version 6.22A, WaveMetrics Inc.). In addition, an improved-ambient elemental analysis was processed by using APES V1.06 (ref. [83]). For mass concentration calculations, the default relative ionization efficiency (RIE) values 1.1, 1.2, 1.3 and 1.4 for nitrate, sulfate, chloride and organics were applied. The RIE for ammonium was 2.65, as determined from the mass-based ionization efficiency calibration. After a comparison to the volume concentration from differential mobility particle sizer measurement, a particle collection efficiency factor of 0.85 was applied to account for the particle losses in the aerodynamic transmission lens and vaporizer.

The monoterpene concentrations at SMEAR II were measured with proton transfer reaction mass spectrometry (PTR-MS) at eight different altitudes (4.2, 8.4, 16.8, 33.6, 50.4, 67.2, 101 and 125 m) according to the procedure described by Taipale et al.[84].

**Statistical methods**. The mean diurnal HOM concentration trends in Fig. 3, average HOM concentrations in Table 1 and average RF$_{ARI}$ values were calculated using the "mean" built in function in MATLAB® R2017b. The Pearson's correlation coefficients (R) in Table 1 were calculated using the "corrcoef" built in function in MATLAB® R2017b. The normalized mean bias (NMB) between the modelled (M) and observed (O) HOM(g) concentrations in Table 1 was calculated using Eq. (7). The median, 25th and 75th percentile particle number concentrations, particle number size distributions and CCN concentrations in Figs. 5 and 7 were calculated using the MATLAB® R2017b built in functions "median" and "prctile" respectively.

$$NMB = \frac{\sum_1^n (M - O)}{\sum_1^n (O)}. \tag{7}$$

## Data availability

All data shown in the figures and tables and additional raw data are available upon request from the corresponding author (P.R.).

## Code availability

The complete PRAM mechanism and all codes used to conduct the analysis presented in this paper can be obtained by contacting the corresponding author (P.R.). The complete PRAM mechanism (Supplementary Tables 4 and 7) written in a format compatible with the Kinetic PreProcessor (KPP)[48] together with all species information listed in Supplementary Tables 2 and 6 can also be downloaded from [https://doi.org/10.1594/PANGAEA.905102]

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

## Acknowledgements

This work was supported by the Swedish Research Council FORMAS (proj. no. 2014-1445, 2015-749, 2018-01745), eSTICC (eScience tools for investigating Climate Change in Northern High Latitudes), the European Commission Horizon 2020 program (grant no. 654109), the European Research Council (grant 638703-COALA, QAPPA-335478), the Swedish Strategic Research Program MERGE, Knut and Alice Wallenberg foundation (academy fellowship AtmoRemove), the Academy of Finland (proj. no. 272041, 299574, 1266388, 1303676), the Centre for Scientific and Technical Computing at Lund University, LUNARC, the CSC IT Center for Science in Espoo, Finland, and the Swedish National Infrastructure for Computing, SNIC. Open access funding provided by Lund University.

## Author contributions

P.R., M.B., M.E., T.K., T.O., M.P.R., D.R.W, T.P, M.K., H.V., I.R. and A.V. designed research; P.R., M.B., M.E., T.K., T.O., M.P.R., C.X., N.S, P.R., L.H., N.H., L.H, J. E. and L.P. performed research; P.R., M.B., E.Ö., J.E., P.C. developed the models; P.R., M.E., N.S, P.R., L.H., N.H. and L.H. analysed data; P.R., M.E., T.K., T.O. and M.B. wrote the paper

## Competing interests

The authors declare no competing interests.
