## [Peer Review File · Nature Communications]

Reviewers' comments:

Reviewer #1 (Remarks to the Author):

The paper by Roldin et al. presents a study on the role of highly oxidated organic molecules (HOMs) and also associated new particle formation (NPF) in affecting cloud (e.g., CCN) and climate (e.g., direct radiation forcing) system over the Boreal forest. HOMs or low-volatility organic compounds have been found previously to play important roles in the aerosol-cloud-radiation system. However they have also been underrepresented in the current atmospheric models due to incomplete understanding. This work is trying to fill in this gap by developing a modeling framework for HOMs and its impacts on aerosol-cloud-climate system and will significantly contribute to the atmosphere/climate community. This paper is well written with great detail of modeling methods and also very interesting analysis. The PRAM model developed in this work coupled with the existing aerosol model (i.e., ADCHEM) can reproduce the measurements well. The radiation forcing (RF) caused by HOM predicted by this work is quite significant when compared to other aerosol species (e.g., sulfate), which indicates its importance to be considered by the atmosphere/climate models. I would definitely recommend it to be accepted after addressing my following minor comments:

Line 108: low O:C ratio compared to what?

Line 140: Table 1, the stats don't seem to be right for a few species here. For example, for HOM RO₂, (0.13-0.10)/0.10=33% instead of 67%. Please double check.

Line 167: Fig. 4, The model simulates wet deposition as well according to Section Atmospheric model simulations in Methods? What's the contribution here or is it ignored in the current simulations?

Line 205: I wouldn't say "accurately" and "moderately" might be more appropriate here, since there are some large underpredictions for the small size of particles (between 10-100 nm) according to Fig5a-b.

Line 283: Fig. S14?

Line 288: by 8-12%

Line 294: contributes

Line 312: Section Summary and Conclusion, would like to see some discussions/implications on how the current model framework could be implemented into 3-D climate/CTM models for more realistic simulations of atmosphere and also feedbacks.

Line 401: how about coagulation?

Line 425: Section Atmospheric model simulations, is aqueous phase chemistry considered?

Line 435: what's the time step?

Line 445&446: due to?

Line 481-482,) is missing

Line 521, Fig.5a?

Reviewer #3 (Remarks to the Author):

Roldin et al present a very comprehensive modelling study where they have simulated using a kinetic Lagrangian parcel model the formation and, using a subsequent cloud parcel model, the impacts of HOMs.

The paper has some key findings which are in good accord with previous literature. Namely that accounting for HOMs has an important impact on climate through new particle formation. This is clearly not a novel result and the novelty of this study is really in the use of a complex chemical scheme. As a basis for their scheme Roldin et al. use the MCMv3.1. The MCM is currently undergoing some major updates to the the RO₂ chemistry and a paper is under review. Some of the conclusions of this review (Jenkin et al., 2019) would be extremely useful to look at in the present study -- for example, I would have expected more formal sensitivity analysis to understand how the use of generic

rate coefficients impacts the results, how uncertainty in the activation energy for the H-shifts affects the vertical profiles of the HOMs etc. There is a deep literature in H-shift reactions (QOOH) which goes back to the 1980s and even earlier. It is very disappointing to see this literature overlooked.

All that said, the paper is generally well written (only a few typos and grammatical errors) and is interesting to read, albeit quite a slog.

Unfortunately, all in all I found that in its present form the manuscript is out of scope for being useful for the wider community as the authors have failed to compare their comprehensive chemical mechanism, clearly a great step forward but absolutely unpractical for actual CTM modellers who are limited to only including a few hundred reactions in total, with the more simple mechanisms proposed earlier and used in actual CTMs (i.e. Gordon et al., 2016, P.N.A.S). For this reason I must recommend rejection and would encourage the authors to include runs with more simplified chemical schemes to better inform the wider community or to perform more runs to really pick out the impacts of uncertainty in key kinetic parameters and submit to a more specialised journal (i.e. ACP). Either way I feel the manuscript falls short.

I think this is a valiant effort and would like to congratulate the author (adding in this amount of chemistry into a model is no simple job I know) and the paper has much promise but I can't support it's publication in this journal.

Point-by-point response to the referees' comments on the manuscript with the following tracking number: NCOMMS-18-15202619A

We thank both reviewers for the comprehensive and constructive review of our manuscript titled "The role of highly oxygenated organic molecules in the Boreal aerosol-cloud-climate system"

If we are informed that the manuscript will be accepted for publication we will upload the complete Peroxy Radical Autoxidation Mechanism (PRAM) as a Kinetic PreProcessor (KPP) compatible input file at the PANGAEA data submission system and add the unique Digital Object Identifier (DOI) and link to the data in the code availability statement.

We updated the code and data availability statements in the manuscript so that it is clear that all source codes and data generated for the manuscript are freely available upon request from me as corresponding author:

Code availability. The complete PRAM mechanism and all codes used to conduct the analysis presented in this paper can be obtained by contacting the corresponding author (P.R.). The complete PRAM mechanism written in a format compatible with the Kinetic PreProcessor (KPP) ⁴⁷ can be downloaded from <https://doi.org/XXX>.

Data availability. All data shown in the figures and tables and additional raw data are available upon request from the corresponding author (P.R.)."

Below you can find our response to all reviewer comments (including the comments from the reviewers). We first answer the comments from Reviewers #1 and then the comments from Reviewers #3.

Reviewer #1 (Remarks to the Author):

The paper by Roldin et al. presents a study on the role of highly oxidated organic molecules (HOMs) and also associated new particle formation (NPF) in affecting cloud (e.g., CCN) and climate (e.g., direct radiation forcing) system over the Boreal forest. HOMs or low-volatility organic compounds have been found previously to play important roles in the aerosol-cloud-radiation system. However they have also been underpresented in the current atmospheric models due to incomplete understanding. This work is trying to fill in this gap by developing a modeling framework for HOMs and its impacts on aerosol-cloud-climate system and will significantly contribute to the atmosphere/climate community. This paper is well written with great detail of modeling methods and also very interesting analysis. The PRAM model developed in this work coupled with the existing aerosol model (i.e., ADCHEM) can reproduce the measurements well. The radiation forcing (RF) caused by HOM predicted by this work is quite significant when compared to other aerosol species (e.g., sulfate), which indicates its importance to be considered by the atmosphere/climate models. I would definitely recommend it to be accepted after addressing my following minor comments:

Thank you.

Line 108: low O:C ratio compared to what?

Good point. We mean low O:C compared to the atmospheric observations and atmospheric modelling results described on Line 238-243 in the manuscript. We suggest that we remove the sentence on Line 108-109 “The modelled HOM SOA has a relatively low O:C because a large fraction of the mass is composed of HOM dimers, which have $O:C \leq 0.6$ (e.g. $C_{20}H_{30}O_{10}$, $C_{20}H_{30}O_{11}$ and $C_{20}H_{30}O_{12}$).” and only discuss the relatively low O:C in the smog chamber experiments compared to the atmospheric observations and modelling on Line 238-243 in the manuscript.

Line 140: Table 1, the stats don't seem to be right for a few species here. For example, for HOM RO₂, (0.13-0.10)/0.10=33% instead of 67%. Please double check.

Thank you for noticing this. The observed average RO₂ concentration is wrong in table 1. It should be 0.08 pptv. We made a mistake and did not calculate the observed average for the complete 10 days but only for the 2 first days. We have double checked all other calculations for table 1 and they should be correct. We have changed the value of the observed average HOM RO₂ concentration to the correct one.

Line 167: Fig. 4, The model simulates wet deposition as well according to Section Atmospheric model simulations in Methods? What's the contribution here or is it ignored in the current simulations?

This is a very relevant question. Wet deposition of HOM(g) were not considered in the model simulations used to generate the results in Fig. 4 or in any other model simulations presented in the manuscript. The wet deposition mentioned in the Atmospheric model simulations method section refer to the removal of aerosol particles, and the in-cloud dissolution of SO₂, H₂O₂, NH₃, HNO₃ and HCHO and oxidation of SO₂ by H₂O₂. We have reformulate the wet deposition statement in the method section from:

“The aerosol dynamics include new particle formation, Brownian coagulation, dry and wet deposition and condensation/evaporation.”

to:

“The aerosol dynamics include new particle formation, Brownian coagulation, condensation/evaporation and dry deposition of particles and gases (including HOM). In and below cloud scavenging of aerosol particles and in-cloud sulfate aerosol formation and scavenging of SO₂, H₂O₂, NH₃, HNO₃ and HCHO were also considered, analogous to Roldin et al.³². However, the model did not consider any cloud droplet aqueous phase chemistry or in- and below cloud scavenging of other organic molecules.”

We have also checked the 1 minute resolution observed precipitation at SMEAR II for the simulated period in May 2013, freely available at <https://avaa.tdata.fi/web/smart>. During the complete 10 day period it was raining in total 14.3 mm of which most (9 mm) came between 10 am and 11:30 am on the 18th of May. For the complete 10-day period, rainfall was observed during only 2.8 % of the time at SMEAR II station. Considering the short lifetime of HOM(g), we expect that even though precipitation may influence the HOM(g) concentration at SMEARII, this will not have any substantial impact of the average HOM(g) concentrations for the selected period.

Line 205: I wouldn't say “accurately” and “moderately” might be more appropriate here, since there are some large underpredictions for the small size of particles (between 10-100 nm) according to Fig5a-b.

We agree that “accurately” is not the most appropriate word to use. We suggest that the change the sentence on line 205-028 from:

“With the CTRL setup, ADCHEM accurately predicts the general trends in the observed particle number size distributions at SMEAR II (Fig. 5a, b) and the magnitude and diurnal trends in the number concentration of particles in the nucleation mode, Aitken mode and accumulation mode (Fig. 5c, d, e).”

To

“With the CTRL setup, ADCHEM predicts the general trends in the observed particle number size distributions at SMEAR II with reasonable accuracy (Fig. 5a, b) and the magnitude and diurnal trends in the number concentration of particles in the nucleation mode, Aitken mode and accumulation mode (Fig. 5c, d, e).”

Line 283: Fig. S14?

Thank you. Yes it should be Fig. S14 and not Fig. S13. We have changed this.

Line 288: by 8-12%

Thank you. We have changed from “with 8-12 %” to “by 8 - 12 %”.

Line 294: contributes

Thank you. We have changed from “HOM contribute” to “HOM contributes”.

Line 312: Section Summary and Conclusion, would like to see some discussions/implications on how the current model framework could be implemented into 3-D climate/CTM models for more realistic simulations of atmosphere and also feedbacks.

We agree with the reviewer that this information is relevant for the large-scale climate/CTM modelling community.

We have added a new result section titled “Recommendations to the atmospheric modelling community” just before the summary and conclusions where we discuss the implication of our work for the large scale atmospheric modelling community, with additional references to the recent publications by McFiggans et al. (2019) and Bianchi et al. (2019). In this section we demonstrate that the comprehensive PRAM mechanism may be substantially reduced if all $RO_2 + RO_2$ reactions that lead HOM dimers are replaced by a small number of reactions that represents how the total pool of RO_2 reacts with the RO_2 in PRAM using single collective rate coefficients. However, we also recommend that PRAM should be evaluated for other regions, e.g. over tropical forests and urban areas before it is implemented and used for any global

scale model applications. We have also added a short final concluding statement about these results in the summary and conclusion section.

All suggested new sections can be found in the end of this document and in the updated manuscript.

New references:

Bianchi et al. Highly Oxygenated Organic Molecules (HOM) from Gas-Phase Autoxidation Involving Peroxy Radicals: A Key Contributor to Atmospheric Aerosol, *Chem. Rev.*, **119**, 6, 3472-3509 (2019).

McFiggans, G. et al. Secondary organic aerosol reduced by mixture of atmospheric vapours, *Nature*, **565**, 587-593 (2019).

Line 401: how about coagulation?

Yes, both codes take into account Brownian coagulation. We have added this to the sentence on Line 400-403:

“They take into account Brownian coagulation and the condensation, dissolution and evaporation of H₂SO₄, NH₃, HNO₃ and all organic oxidation products from Master Chemical Mechanism version 3.3.1 (MCM v.3.3.1) and from the PRAM mechanism (SI Table S2) with pure liquid saturation vapour pressures (p_0) less than 10⁻² Pa (in total 828 species at 290 K).”

Line 425: Section Atmospheric model simulations, is aqueous phase chemistry considered?

Yes for some inorganic compounds but not for organics. For aerosol particles ADCHEM considers the condensation, dissolution and evaporation of sulfuric acid, ammonia, nitric acid, hydrochloric acid and different organic compounds using the analytic prediction of condensation (APC) scheme and predictor of non-equilibrium growth (PNG) scheme developed and described in detail by Jacobson (1997, 2005). The inorganic aerosol equilibrium liquid water content and the effective Henry's law coefficients for HNO₃, HCl and NH₃ is updated for each 30 seconds model time step before the condensation algorithm is called.

In-cloud sulfate aerosol formation and scavenging of SO₂, H₂O₂, NH₃, HNO₃ and HCHO were considered in grid cells where the RH was larger or equal to 98 %, similar to (Roldin et al., 2011). The clouds were assumed to have a fixed supersaturation (S) of 0.2 % and liquid water content (LWC) of 0.5 g m⁻³. The critical supersaturation (Sc) required to activate particles in each size bin were calculated using the analytic formula for the critical supersaturations derived by Kokkola et al. (2008). This formula, which is based on Köhler theory, takes into account the impact of insoluble inclusions in the CCN. In this work, the soot particle content was the only material that was assumed to be present as an insoluble spherical inclusions in the droplets. The organic mass fraction was assumed to be fully water soluble at the critical supersaturation and inorganic salts fully dissociated. The corresponding single droplet mass for each activated aerosol particle (Sc < S) were calculated by dividing the LWC with the total number of activated particles, assuming that all activated cloud droplets had the same size. The dissolution of SO₂ and H₂O₂ and formation of sulphate aerosol mass were considered as described in Roldin et al. (2011).

Jacobson, M. Z.: Numerical techniques to solve condensational and dissolutional growth equations when growth is coupled to reversible aqueous reactions, *Aerosol Sci. Technol.*, **27**, 491–498, 1997.

Jacobson, M. Z.: A Solution to the Problem of Nonequilibrium Acid/Base Gas-Particle Transfer at Long Time Step, *Aerosol Sci. Technol.*, **39**, 92–103, 2005.

Kokkola, H., Vesterinen, M., Anttila, T., Laaksonen, A. and Lehtinen, K. E. J.: Technical note: Analytical formulae for the critical supersaturations and droplet diameters of CCN containing insoluble material, *Atmos. Chem. Phys.*, **8**(7), 1985–1988, doi:10.5194/acp-8-1985-

2008, 2008.

Roldin, P., Swietlicki, E., Schurgers, G., Arneth, A., Lehtinen, K. E. J., Boy, M. and Kulmala, M.: Development and evaluation of the aerosol dynamics and gas phase chemistry model ADCHEM, Atmos. Chem. Phys., 11, 5867–5896, doi:10.5194/acp-11-5867-2011, 2011.

Based on your previous comment concerning the wet deposition we updated Atmospheric model simulations method section with the following sentences:

“The aerosol dynamics include new particle formation, Brownian coagulation, condensation/evaporation and dry deposition of particles and gases (including HOM). In and below cloud scavenging of aerosol particles and in-cloud sulfate aerosol formation and scavenging of SO₂, H₂O₂, NH₃, HNO₃ and HCHO were also considered, analogous to Roldin et al.³². However, the model did not consider any cloud droplet aqueous phase chemistry or in- and below cloud scavenging of other organic molecules.”

Line 435: what’s the time step?

ADCHEM use a main model time step of 30 seconds. ACDC is called for every time step and updates the molecule cluster distribution. ACDC keeping track of the previous time step molecule cluster distribution (concentrations of molecule clusters of different size and composition).

We have added the following sentence as the second sentence in the Atmosphere Modelling section

“The main model time step was 30 seconds.”

Line 445&446: due to?

Yes, thank you. We have changed from “due” to “due to”

Line 481-482,) is missing

Thank you. We have changed this.

Line 521, Fig.5a?

Yes, thank you. We have changed to Fig. 5a.

Reviewer #3 (Remarks to the Author):

“Roldin et al present a very comprehensive modelling study where they have simulated using a kinetic Lagrangian parcel model the formation and, using a subsequent cloud parcel model, the impacts of HOMs.”

Thank you.

“The paper has some key findings which are in good accord with previous literature. Namely that accounting for HOMs has an important impact on climate through new particle formation. This is clearly not a novel result and the novelty of this study is really in the use of a complex chemical scheme. As a basis for their scheme Roldin et al. use the MCMv3.1. The MCM is currently undergoing some major updates to the the RO2 chemistry and a paper is under review. Some of the conclusions of this review (Jenkin et al., 2019) would be extremely useful to look at in the present study -- for example, I would have expected more formal sensitivity analysis to understand how the use of generic rate coefficients impacts the results, how uncertainty in the activation energy for the H-shifts affects the vertical profiles of the HOMs etc. There is a deep literature in H-shift reactions (QOOH) which goes back to the 1980s and even earlier. It is very disappointing to see this literature overlooked.”

Just to clarify, we have used the presently latest version of MCM (MCMv3.3.1).

When we submitted our manuscript to Nature Communications in November 2018 the Jenkin et al., 2019 paper was not even submitted to ACPD so we had no chance to take this study into account. At present (2019-07-15) the latest freely available version of MCM is still v3.3.1 (<http://mcm.leeds.ac.uk/MCM/>). This is also the version we have used for all model simulations. When a new versions of MCM become publically available we will implement them together with PRAM in ADCHAM and ADCHEM. However, this is not possible yet. We agree that this is a relevant review and we refer to this paper in the updated manuscript. We do this in the new extended method section titled “PRAM sensitivity analysis”, which include a more formal sensitivity analysis of the RO2+RO2 rate coefficients and the activation energy for the H-shifts.

In PRAM we use the generic rate coefficient for non-acyl peroxy radicals reacting with NO as is presently implemented in MCMv3.3.1. This is the same expression and as suggested by Jenkin et al., 2019:

$$K_{RO2NO} = 2.7E-12 * EXP(360/T) \quad (Eq. 1)$$

Jenkin et al., 2019 propose a new generic rate coefficient expression for RO₂ + HO₂ reactions of non-acyl peroxy radicals:

$$K_{RO2HO2} = 2.8E-13 * EXP(1300/T) * EXP([1 - 0.23 * n_{CON}]) \quad (Eq. 2)$$

Here n_{CON} stand for the number of carbon, oxygen and nitrogen atoms in the organic group (R) of the peroxy radical (i.e. excluding the peroxy radical oxygen atoms and equivalent to the carbon number in alkyl peroxy radicals).

In PRAM we have used the generic rate coefficient as presently is implemented used by MCMv3.3.1:

$$K_{RO2HO2} = 2.91E-13 * EXP(1300/T) \quad (Eq. 3)$$

The temperature dependence of these two expressions for the RO₂ + HO₂ reaction rate coefficients are the same. Also since PRAM considers RO₂ with 10 C atoms and at least 2 O atoms in the organic group (e.g. C₁₀H₁₅O₄) n_{CON} is in our case are always larger or equal to 12. At this lower limit of n_{CON} in PRAM Eq. 2 gives 25 % higher K_{RO2HO2} than Eq. 1 while for RO₂ with $n_{CON}=20$ e.g. C₁₀H₁₅O₁₂ the difference is only 7 %. To be consistent with the presently most recent publically available version of MCM (MCMv3.3.1) we do not intent to change the K_{RO2HO2} reaction rates in PRAM for this study.

In the present version of MCMv3.3.1 the generic reaction rate constants for $RO_2 + RO_2$ reactions range from $10^{-11} \text{ cm}^3 \text{ molecule}^{-1} \text{ s}^{-1}$ to $6.7 \cdot 10^{-15} \text{ cm}^3 \text{ molecule}^{-1} \text{ s}^{-1}$ for acyl peroxy radicals and tertiary carbon peroxy radicals, respectively. These rate coefficients are based on reaction rates of generally less oxygenated and smaller RO_2 molecules than the RO_2 in PRAM. In PRAM we have used $RO_2 + RO_2$ reaction rates leading to closed shell monomers in the range of $5 \cdot 10^{-12} \text{ cm}^3 \text{ molecule}^{-1} \text{ s}^{-1}$ to $10^{-11} \text{ cm}^3 \text{ molecule}^{-1} \text{ s}^{-1}$ and $RO_2 + RO_2$ reaction rates leading to closed shell HOM dimers ranging between $10^{-13} \text{ cm}^3 \text{ molecule}^{-1} \text{ s}^{-1}$ to $5 \cdot 10^{-12} \text{ cm}^3 \text{ molecule}^{-1} \text{ s}^{-1}$, with the lowest values applied to the least oxygenated RO_2 and the highest values used for the most oxygenated RO_2 (See Table S4). Presently we do not know the exact molecular structure of each individual HOM RO_2 species formed upon oxidation of monoterpenes and each observed mass peak are likely representing the total concentration of a several RO_2 isomers. Hence, we are aware of that the $RO_2 + RO_2$ reaction rates of the individual RO_2 are uncertain. The relatively high $RO_2 + RO_2$ reaction rates needed in PRAM for satisfactory representation of the observations in the JPAC experiments most likely reflect the multi-functionality of the HOM RO_2 . With the applied reaction rate coefficients as tabulated in Table S4, PRAM captures the observed trend in the HOM RO_2 concentrations in JPAC as a function of the α -pinene reaction rate with O_3 (Fig. 1c) and the trends and absolute concentrations of the RO_2 in the atmosphere reasonably well (Fig. 3a). We have added new figures and a table to the supplement where we compare the observed and modelled HOM concentrations in JPAC and at SMEAR II when we scale all $RO_2 + RO_2$ reaction rates in PRAM (Table S4) up or down with a factor of 2 (see Fig. S15-S18 and a Table S9 in the updated supplementary material).

Concerning the activation energy for the H-shifts reactions, which are rate limiting for the RO_2 autoxidation, we decided to use an activation energy within the range of values suggested by Rissanen et al. (2015). Rissanen et al. (2015) calculated activation energies in the range of 90 - 120 kJ for RO_2 isomers formed from ozonolysis of α -pinene, which is higher than the activation energies calculated by Rissanen et al. (2014) for cyclohexene (70-80 kJ). In PRAM we have used an activation energy of 100 kJ for all H-shift reactions of RO_2 formed from the oxidation of monoterpenes. However we have added references to Rissanen et al. (2014), Praske et al. (2019) and Jenkin et al. (2019) as additional publications investigating the temperature dependence of the H-shift reactions of RO_2 formed from other VOCs.

According to Praske et al. (2018) the RO_2 H-shifts of three different RO_2 formed from OH-oxidation of 2-hexanol increases with factors in the range of 4.3 to 7 between 296 K and 318 K. This correspond to activation energies in the range of 50 kJ to 70 kJ.

Jenkin et al. (2019) list some published H-shift isomerization reaction rates of peroxy radicals. Most of these have activation energies in the same range as measured by Praske et al. (2018) and calculated for cyclohexene by Rissanen et al. (2014), i.e. 40 kJ to 80 kJ.

Since the H-shift activation energies from these other studies are up to about a factor of 2 lower than the values proposed by Rissanen et al. (2015) for RO_2 formed from α -pinene ozonolysis, we have now also performed additional smog chamber and atmospheric model sensitivity tests using a factor two lower activation energies for the H-shift reactions (i.e. 50 kJ), but making sure that the H-shift reaction rates always are the same at the reference temperature of the JPAC experiments (i.e. 289.15 K). In addition, we have also performed model sensitivity tests using a H-shift activation energy of 150 kJ. This sensitivity test with higher activation energies is motivated by that Qu  l  ver et al. (2019) recently showed that the observed HOM yields during α -pinene ozonolysis experiments (using initial concentrations of 50 ppb α -pinene and 100 ppb O_3) is about 50 times lower at 273.15 K compared to 293.15 K, which most likely requires a very strong temperature dependence at least for the first and second H-shift reaction rates.

Figure S18 in the new supplementary material illustrates how different values of the activation energy for the RO₂ H-shifts affects the average vertical HOM concentration profiles at SMEAR II for the simulated period in May 2013.

New references:

Praske, E., et al. Atmospheric autoxidation is increasingly important in urban and suburban North America. *Proc. Natl. Acad. Sci. U. S. A.*, **115**, 64-69 (2018).

Jenkin, M. E., Valorso, R., Aumont, B., & Rickard, A. R. Estimation of rate coefficients and branching ratios for reactions of organic peroxy radicals for use in automated mechanism construction, *Atmos. Chem. Phys.*, **19**, 7691-7717 (2019).

Quéléver, L. L. J. et al. Effect of temperature on the formation of highly oxygenated organic molecules (HOMs) from alpha-pinene ozonolysis, *Atmos. Chem. Phys.*, **19**, 7609-7625 (2019).

“All that said, the paper is generally well written (only a few typos and grammatical errors) and is interesting to read, albeit quite a slog.

Unfortunately, all in all I found that in its present form the manuscript is out of scope for being useful for the wider community as the authors have failed to compare their comprehensive chemical mechanism, clearly a great step forward but absolutely unpractical for actual CTM modellers who are limited to only including a few hundred reactions in total, with the more simple mechanisms proposed earlier and used in actual CTMs (i.e. Gordon et al., 2016, P.N.A.S). For this reason I must recommend rejection and would encourage the authors to include runs with more simplified chemical schemes to better inform the wider community or to perform more runs to really pick out the impacts of uncertainty in key kinetic parameters and submit to a more specialised journal (i.e. ACP). Either way I feel the manuscript falls short.

I think this is a valiant effort and would like to congratulate the author (adding in this amount of chemistry into a model is no simple job I know) and the paper has much promise but I can't support it's publication in this journal.”

We do not agree with the reviewer #3 that the manuscript is out of scope for being useful for the wider community. We do not know exactly what reviewer #3 mean with “the wider community”. Possibly it is global scale modelers. A fundamental understanding of the atmospheric processes that governs the formation and lifetime of HOM and new particles in the atmosphere must be of interest to a wide atmospheric community, both for modelers and experimentalists. Without the fundamental process knowledge we cannot know if the models predicts the “correct” results for the right or wrong reasons.

This is supported by the recent comprehensive review on HOM by Bianchi et al. (2019) were you e.g. can find the following statement:

“... to constrain the contribution of HOM to the growth of particles in the atmosphere, methods are needed that combine both observations and models that can provide detailed information about the gas-phase chemistry, HOM volatility (i.e., p_0 and reactivity in the condensed phase), aerosol dynamics, dry deposition losses, and the planetary boundary layer mixing. To our knowledge, there exist no published studies that provide a complete closure of the modeled and observed HOM(g), the particle composition, and growth rates.”

Also in the recent Nature paper by McFiggans et al. (2019) is stated in the abstract that the “formation mechanisms of secondary organic aerosol in the atmosphere need to be considered more realistically, accounting for mechanistic interactions between the products of oxidizing precursor molecules”

McFiggans et al. (2019) point out the complex effects on SOA mass yields when different VOCs are mixed in the atmosphere and that tabulated SOA mass yields from individual VOC precursor experiments should be used with caution when modelling aerosol formation. **“Measuring SOA yields from individual compounds leads to insight into the mechanisms of SOA production but the results do not reflect the conditions of the real environment. Such data should therefore be used with caution when modelling aerosol formation. In the general case, the amounts of HOMs, HOM-RO₂ and RO₂ products from potential SOA precursors as well as from volatile compounds that do not produce SOA mass should be considered when predicting the mixture’s yield.”**

In the present study we use methods that address these concerns and hence, we do think that our manuscript is very relevant for the wider atmosphere research community.

We agree with reviewer #3 and reviewer #1 that it is relevant and important to provide concrete suggestions for how the presented results can be used for the wider large scale 3D-CTM community. In line with McFiggans et al. (2019), our recommendations are that CTMs should at least aim to use near-explicit representations of HOM formation, which can represent the non-linear effects caused by e.g. temperature, NO_x and RO₂ chemistry. However, for the aerosol particle growth the formed closed shell HOM species may be lumped into e.g. a volatility basis set. Both the effects of varying RO₂ concentrations and temperatures for the HOM formation will need further investigation.

For the HOM dimers formation, PRAM explicitly simulates the individual RO₂ + RO₂ reactions between 94 RO₂ from the MCMv3.3.1 chemistry and 17 RO₂ in PRAM. This results in 1598 reactions. However, instead of using an explicit representation of individual HOM dimer formation between individual RO₂ from the MCM chemistry and RO₂ in PRAM, the dimer formation may be parameterized assuming that the total pool of RO₂ react with the individual RO₂ in PRAM, similar to how the mechanism represent the RO₂ + RO₂ reactions leading to closed shell HOM monomers. This, drastically reduces the total number of reaction in PRAM from 1773 to 192 and the number of species from 208 to 89. In the reduced PRAM version the HOM dimers are only represented by two dimers, one for the HOM formed from monoterpene ozonolysis and one for dimers formed from OH-oxidation of monoterpenes. In this case the formed HOM dimers will have to be represented with some average properties of typical HOM dimers, e.g. molar mass and p_0 . However, since the majority of HOM dimers formed from monoterpenes are ELVOCs which condenses irreversible to the existing aerosol particles, this simplification introduces very minor deviation in the modelled SOA mass formation. In the new manuscript we have included model results from SMEAR II when we test this simplified representation of the closed shell dimer formation in PRAM. In JPAC, the modelled concentrations of closed shell HOM monomers, dimers, HOM RO₂ and SOA are almost identical with the full and reduced PRAM mechanism. This, is not surprising since all RO₂ in the JPAC experiments were formed from α -pinene as the single precursor. However, also for the atmospheric model simulations at SMEAR II the model HOM(g) concentrations are very similar with the reduced and full PRAM mechanism (see the new Fig. S17 and Table S9). Partly this may be reflected by that the dominant source of RO₂ at SMEAR II are monoterpenes. For future studies, we recommend that PRAM should be evaluated also for conditions where the majority of atmospheric RO₂ pool are originating from precursors that do not contribute substantially to HOM formation, e.g. isoprene.

New references:

Bianchi et al. Highly Oxygenated Organic Molecules (HOM) from Gas-Phase Autoxidation Involving Peroxy Radicals: A Key Contributor to Atmospheric Aerosol, *Chem. Rev.*, **119**, 6, 3472-3509 (2019).
McFiggans, G. et al. Secondary organic aerosol reduced by mixture of atmospheric vapours, *Nature*, **565**, 587-593 (2019).

Based on the comments from reviewer #1 and #3 the following sections has been added to the new manuscript on lines 313-342, 366-369 and 433-485:

“Recommendations to the atmospheric modelling community. Climate and chemistry transport models need to represent the formation and losses of HOM more realistically in order to improve the predictions of SOA formation and its implications for air quality and climate on Earth^{1,18,19}.

In this work, we have developed and used the near explicit Peroxy Radical Autoxidation Mechanism (PRAM) to provide a complete closure between the modelled and observed HOM concentrations at the Boreal forest station SMEAR II. This fundamental process knowledge is required to efficiently improve the representation of HOM SOA formation in atmospheric models. However, for most large scale atmospheric model applications, PRAM and the SOA formation scheme used in this work may need to be reduced (simplified). Still, it is important that any reduced mechanism should be able to represent the non-linear SOA yield effects caused by different O₃, OH, RO₂, NO_x and temperature conditions. If the gas-phase chemistry mechanism (e.g. PRAM) can fulfill these requirements, the particle growth and/or SOA mass formation may be successfully parameterized by lumping the formed closed shell species into a volatility basis set framework^{18,23,33}. This can reducing the number of condensable compounds from several hundreds to less than ten.

PRAM explicitly treats RO₂ + RO₂ reactions between 94 RO₂ from the MCMv3.3.1 chemistry and 17 PRAM specific RO₂ (Table S4), in total 1598 reactions. However, instead of representing the reactions between individual RO₂ the dimer formation may be parameterized assuming that the total concentration of RO₂ in MCMv3.3.1 (the so called RO₂ “pool”)⁷ react with the RO₂ in PRAM using single collective rate coefficients. If the formed HOM dimers are only represented by two dimers, one for the HOM formed from monoterpene ozonolysis and one for dimers formed from OH-oxidation of monoterpenes the total number of reaction in PRAM is reduced from 1773 to 192 and the total number of species from 208 to 89 (see Table S8). In this case the formed HOM dimers will have to be represented with some average properties of typical HOM dimers, e.g. molar mass and p_0 . However, since the majority of HOM dimers formed from monoterpenes are ELVOCs, which condenses irreversible to the existing aerosol particles, this simplification can be acceptable from a SOA mass formation perspective. At SMEAR II the reduced PRAM version (Table S8) gives almost identical average total HOM concentrations and only 6 % higher average HOM dimer concentrations compared to the default PRAM version (Table S4) (Fig. S17, Table S9). The

close agreement between the two PRAM versions and the observations at SMEAR II is partly reflected by that the dominant source of RO₂ at SMEAR II are the locally emitted monoterpenes. Thus, in order to conclude about the applicability of the reduced and full PRAM versions for global scale model applications, they should be evaluated also for conditions where a major fraction of the RO₂ pool is originating from precursors that do not contribute substantially to HOM formation, e.g. in environments with high isoprene concentrations.”

This statement has been added to the summary and conclusions (line 366-369):

“We demonstrate that the comprehensive PRAM mechanism may be substantially reduced. The reduced PRAM version can likely be used for realistic representations of HOM SOA formation in regional and global scale CTMs. However, before PRAM is used for large scale CTM applications we recommend that the mechanism is evaluated also for other regions, e.g. over tropical forests and urban areas. “

These sections have been added to the Methods (line 433-485):

PRAM sensitivity analysis. In MCMv3.3.1 the generic reaction rate constants for RO₂ + RO₂ reactions range from 10⁻¹¹ cm³ molecule⁻¹ s⁻¹ for acyl peroxy radicals to 6.7·10⁻¹⁵ cm³ molecule⁻¹ s⁻¹ for tertiary carbon peroxy radicals^{7,24,25}. These rate coefficients are based on measured reaction rates of generally less oxygenated and smaller RO₂ molecules than the RO₂ in PRAM. PRAM uses RO₂ + RO₂ reaction rates leading to closed shell monomers in the range of 5·10⁻¹² cm³ molecule⁻¹ s⁻¹ to 10⁻¹¹ cm³ molecule⁻¹ s⁻¹ and RO₂ + RO₂ reaction rates leading to closed shell HOM dimers between 10⁻¹³ cm³ molecule⁻¹ s⁻¹ to 5·10⁻¹² cm³ molecule⁻¹ s⁻¹, with the lowest values applied to the reactions involving the least oxygenated peroxy radicals and the highest values for the most oxygenated peroxy radicals (see Table S4). With the reaction rate coefficients as tabulated in Table S4, PRAM match the observed HOM RO₂ concentrations in JPAC (Fig. 1c) and the trends and absolute concentrations of HOM RO₂ in the atmosphere reasonably well (Fig. 3a). Figure S15 compares the observed and modelled HOM concentrations in JPAC when we scale all RO₂ + RO₂ reaction rates (k(RO₂ + RO₂)) in PRAM up or down with a factor of two. With k(RO₂ + RO₂)x0.5, the lifetime and concentration of RO₂ increases. At an atmospheric relevant α-pinene + O₃ reaction rate of 0.3 ppt_v s⁻¹ and k(RO₂ + RO₂)x0.5 the HOM RO₂ concentrations become 60 % higher, while with k(RO₂ + RO₂)x2 the modelled HOM RO₂ become 40 % lower. At low α-pinene + O₃ reaction rates the modelled closed shell HOM formation is limited by the formation of highly oxygenated RO₂ via autoxidation and by the bimolecular termination reactions that lead to closed shell products. At conditions with high α-pinene + O₃ reaction rates (high absolute RO₂ concentrations) the closed shell HOM formation are primarily limited by the formation of the highly oxygenated RO₂ and not by the bimolecular termination reactions. Instead, RO₂ + RO₂ reactions can cause termination of the autoxidation reaction chain before many of the RO₂ become highly oxygenated. Thus, in the low RO₂ concentration regime, higher k(RO₂ + RO₂) results in higher closed shell HOM concentrations, while in the high RO₂ concentration regime, higher k(RO₂ + RO₂) results in lower closed shell HOM concentrations. This is why the modelled closed shell HOM concentrations (monomers and dimers) are slightly higher at α-pinene + O₃ reaction rates < 0.4 ppt_v s⁻¹ but lower at α-pinene + O₃ reaction rates > 0.4 ppt_v s⁻¹ in the k(RO₂ + RO₂)x2 run compared to default PRAM setup (k(RO₂ + RO₂)x1). The opposite trends can be seen for the k(RO₂ + RO₂)x0.5 simulation. The HOM observations in JPAC indicate that the absolute closed shell HOM monomer and dimer yields decreases somewhat when the α-pinene + O₃ reaction rates increases in the chamber. These results are consistent with the PRAM model simulations which uses k(RO₂ + RO₂)x2. However, at

atmospheric relevant α -pinene + O₃ reaction rates < 0.5 ppt_v s⁻¹ the modelled closed shell HOM concentrations are relatively insensitive to the exact values of k(RO₂ + RO₂) (Fig. S15).

The temperature dependence of the autoxidation reaction rates in PRAM (Table S4) all corresponds to an activation energy of 100 kJ for the rate limiting H-shifts ($E_{H-shift}$). This activation energy is within the range of values suggested by Rissanen et al.¹⁴, which calculated $E_{H-shift}$ in the range of 90 kJ - 120 kJ for different RO₂ isomers formed from ozonolysis of α -pinene. The H-shift activation energy used as default in PRAM is higher than the $E_{H-shift}$ measured and calculated for peroxy radicals originating from several other VOCs, which generally are in the range 40 kJ to 80 kJ^{11,49,50}. However, Quéléver et al.⁵¹ recently showed that the observed HOM yields during α -pinene ozonolysis experiments in the AURA chamber was about 50 times lower at 273 K compared to 293 K. The AURA experiments were performed using an initial α -pinene and O₃ concentrations of 50 ppb and 100 ppb respectively. This corresponds to an α -pinene + O₃ reaction rate of ~10 ppt_v s⁻¹. The results from Quéléver et al. indicate that the autoxidation reaction rates of RO₂ formed from α -pinene ozonolysis must slow down considerably between 293 K and 273 K. This, together with the presumably high absolute RO₂ in the AURA experiments (i.e. short lifetime of RO₂ with respect to RO₂ + RO₂ reactions) may at least partly explain the observed drastic drop in the HOM yield between 293 K and 273 K⁵¹. With the default $E_{H-shift}$ of 100 kJ in PRAM, the modelled HOM molar yield at an α -pinene + O₃ reaction rate of ~1 ppt_v s⁻¹ increases from 2.3 % to 9.0 % between 270 K and 310 K, while with $E_{H-shift} = 50$ kJ the yield range between 4.4 % and 8.3 % and with $E_{H-shift} = 150$ kJ the HOM molar yields range between 1.6 % and 9.0 % (Fig. S16). For all model sensitivity tests the absolute autoxidation reaction rates were kept identical at the reference temperature 289 K, i.e. the same temperature as was used during the JPAC experiments⁹.

In Fig. S17 and Table S9 we compare the modelled HOM concentrations with the observations at SMEAR II for different model sensitivity tests where we scaled all RO₂ + RO₂ reaction rates in PRAM up or down with a factor of two, or change the activation energy of the H-shift reaction rates from the default 100 kJ to 50 kJ or 150 kJ respectively. The differences in the modelled total HOM concentrations between the different model simulations are relatively small (e.g. FAC2 values between 0.92 and 0.94). Fig. S18 compares the average vertical HOM concentration profiles at SMEAR II from the different PRAM sensitivity tests. All concentration profiles are within ± 10 % from the default PRAM model simulation results at all altitudes. Thus, the modelled HOM concentrations for the simulated period at SMEAR II seem to be relatively robust, considering the estimated range of uncertainty in the absolute RO₂ + RO₂ reaction rates and the H-shift activation energies. The small differences between the model sensitivity tests with different values of $E_{H-shift}$ is related to that the average surface temperatures at SMEAR II were 287.9 K for the simulated period.

Reduced PRAM version. In order to be able to implement PRAM into large-scale CTMs the number of reactions and species need to be minimized. Instead of considering reactions between individual RO₂ that form HOM dimers (R85-R1118 and R1193-R1756, in Table S4), it may adequate and necessary to be represent these reactions using a simplified approach where the total pool of RO₂ are allowed to react with the individual RO₂ in PRAM using single collective rate coefficients (Table S8, R85-R95 and R170-R175). This, drastically reduces the total number of reaction in PRAM from 1773 to 192 and the number of species from 208 to 89. In the JPAC experiments, where all RO₂ are originating from α -pinene, this simplification introduces no noticeable model deviation concerning the total HOM gas-phase concentrations or the modelled SOA formation.”

In addition to the manuscript changes related to the review comments we have made the following modifications to the manuscript or supplement:

- *Co-author Noora Hyttinen's affiliation was changed.*
- *The Swedish Research Council FORMAS proj. no. 2018-01745 is added to the acknowledgement*
- *The estimated Henry's law coefficients from COSMOTerm was updated in Table S6 in the supplement. It was a mistake in the original calculations when picking conformers that had less intramolecular hydrogen bonds. The experimental data for H₂O, which is in the COSMOTerm database was also added to the new calculation in order to make it more accurate. The new coefficients calculated using all conformers also changed a little, but the changes are very small.*

REVIEWERS' COMMENTS:

Reviewer #1 (Remarks to the Author):

Thanks for the revised version. I think the authors have fully addressed my previous comments especially for the new section by discussing the potential of a reduced version of model to the atmospheric modeling community. I only have a couple of minor editorial corrections as below. I recommended the paper to be accepted once they are addressed.

- 1) line 324, reducing to reduce
- 2) line 329, reacts

Reviewer #3 (Remarks to the Author):

Dear Dr Roldin,

Thank you for addressing the comments that I made on your original manuscript.

I feel that you have done an excellent job in responding to these comments and I will recommend to the editor that your revised manuscript should be published as is.

On a very minor side note, I would urge you and your team to think about the sustainable development of the new mechanism, for example a living repository (on git) that others can work with rather than a doi for some archived code. Down the line, I think there is probably the need for a community wide effort to try and understand how different approaches to simulating the complex chemistry of HOM formation can alias results from numerical models. And I think near explicit mechanisms like yours should act as the standards with which simpler mechanisms can be tested against and optimised for.

All in all, I think the current study is a great step in the right direction and will get many global scale modellers thinking.

Best wishes,
Dr Alexander T. Archibald

Point-by-point response to the referees' comments on the manuscript with the following tracking number: NCOMMS-18-15202619A

We thank both reviewers for that they were willing to review the manuscript a second time and for the kind words and recommendations to accept the paper.

Reviewer #1 (Remarks to the Author):

Thanks for the revised version. I think the authors have fully addressed my previous comments especially for the new section by discussing the potential of a reduced version of model to the atmospheric modeling community. I only have a couple of minor editorial corrections as below. I recommended the paper to be accepted once they are addressed.

- 1) line 324, reducing to reduce
- 2) line 329, reacts

Thank you. We have corrected these typos

Reviewer #3 (Remarks to the Author):

Dear Dr Roldin,

Thank you for addressing the comments that I made on your original manuscript.

I feel that you have done an excellent job in responding to these comments and I will recommend to the editor that your revised manuscript should be published as is.

On a very minor side note, I would urge you and your team to think about the sustainable development of the new mechanism, for example a living repository (on git) that others can work with rather than a doi for some archived code. Down the line, I think there is probably the need for a community wide effort to try and understand how different approaches to simulating the complex chemistry of HOM formation can alias results from numerical models. And I think near explicit mechanisms like yours should act as the standards with which simpler mechanisms can be tested against and optimised for.

All in all, I think the current study is a great step in the right direction and will get many global scale modellers thinking.

Best wishes,

Dr Alexander T. Archibald

Thank you for these recommendations and kind words Alexander. We also think it is a very good idea to create a living repository (e.g. on git). I will discuss this idea with my colleagues which have more experience in using git repositories. Hopefully we can create such a repository soon, but not within the timeframe of this paper. To make our method transparent and reproducible we also think it is important to add the present PRAM code version on an open archive and refer to it using a doi.